

# Josephson oscillations in split one-dimensional Bose gases

**Yuri D. van Nieuwkerk[1⋆], Jörg Schmiedmayer[2] and Fabian H.L. Essler[1]**

**1** Rudolf Peierls Centre for Theoretical Physics, Parks Road, Oxford OX1 3PU
**2** Vienna Center for Quantum Science and Technology (VCQ), Atominstitut,
TU-Wien, Vienna, Austria

⋆ yuridaniel@gmail.com

## Abstract

We consider the non-equilibrium dynamics of a weakly interacting Bose gas tightly confined to a highly elongated double well potential. We use a self-consistent time-dependent Hartree–Fock approximation in combination with a projection of the full three-dimensional theory to several coupled one-dimensional channels. This allows us to model the time-dependent splitting and phase imprinting of a gas initially confined to a single quasi one-dimensional potential well and obtain a microscopic description of the ensuing damped Josephson oscillations.

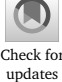

# 1 Introduction

Over the last decade and a half quasi-one-dimensional Bose gases have provided a key platform for experimental studies of non-equilibrium evolution in isolated one-dimensional many-particle quantum systems, see e.g. [1–16]. This has been an important driver of intense theoretical activities aimed at understanding non-equilibrium dynamics in paradigmatic models in $D = 1$, see [17–22] for recent reviews on the subject. A very nice aspect of the cold atom experiments is that they provide quantum simulators for low dimensional quantum field theories. For example trapped single-component Bose gases are well described [23] by the Lieb-Liniger model [24] of a non-relativistic complex scalar field. At low energy densities effective field theory descriptions [25, 26] can apply surprisingly well even out of equilibrium [27–30]. This has led to novel applications of Luttinger liquid theory and its variants such as the study of full distribution functions of quantum observables [31–40]. By modifying the experimental setups it is in principle possible to engineer particular perturbations to Luttinger liquid theory. An important example is given by a system of two tunnel-coupled repulsive Bose gases [41–44], which gives rise to a low-energy description in terms of a Luttinger liquid and a sine-Gordon quantum field theory [45]. The sine-Gordon model is a paradigmatic relativistic quantum field theory that has attracted a huge amount of attention over the last four decades. It has the attractive feature that it is exactly solvable [46–48] and has a number of known applications in the solid state context, see e.g. [26, 49–53]. Motivated by the experimental realization via two tunnel-coupled Bose gases the non-equilibrium dynamics of the sine-Gordon model has been explored by a number of groups and methods [54–66].

Given that the sine-Gordon description only applies in an appropriate scaling limit a crucial question is how close the experiments are to this regime. In equilibrium correlation functions obtained from time-of-flight measurements of the boson density were found to be in good agreement with classical field simulations of the sine-Gordon model [11]. In non-equilibrium situations like the ones studied in Refs. [10, 14, 15] the situation is much less clear. In these experiments two elongated Bose gases are prepared in a quantum state characterized by a phase difference between the two gases. A tunnel coupling between the gases is then applied, which induces Josephson-like oscillations of density and phase. These oscillations quickly damp out and the distribution function of the phase is seen to narrow. Various studies based on the sine-Gordon model have so far failed to account for these observations [62, 65, 66]. In particular, taking into account Gaussian fluctuations on top of the solution of the classical field equations in a self-consistent way produces only very weakly damped Josephson-like oscillations [62, 66]. Given this state of affairs it is natural to question whether the experiments are in the right regime for a sine-Gordon based description to apply. In the experiments one

deals with three dimensional bosons in a time-dependent confining potential. An obvious question is how good the low-energy projection to two one-dimensional Bose gases is in the experimentally relevant parameter regime. Another important issue is that the initial state that is prepared after splitting the gas and imprinting a phase difference is in fact not known, as the splitting process has so far only been modelled in a qualitative phenomenological way [67,68], or via methods that rely on a two-mode approximation [69], a classical field approximation [70] or a restriction to the transverse direction only [69]. In order to start addressing these questions we return to the drawing board and consider a gas of weakly interacting bosons subject to a tight harmonic potential in the $z$-direction, a time-dependent double well potential $V_\perp(y, t)$ in the $y$-direction and a shallow harmonic potential in the $x$-direction. This leads to the Hamiltonian

$$H_{3D}(t) = \int d^3\vec{z}\ \hat{\Psi}^\dagger(\vec{z})\left[-\frac{\nabla^2}{2m} + \frac{m\omega_x^2}{2}x^2 + V_\perp(y, t) + \frac{m\omega_z^2}{2}z^2\right]\hat{\Psi}(\vec{z})$$
$$+ \frac{1}{2}\int d^3\vec{z}\ d^3\vec{z}\,'\hat{\Psi}^\dagger(\vec{z}\,')\hat{\Psi}^\dagger(\vec{z})\hat{U}(\vec{z} - \vec{z}\,')\hat{\Psi}(\vec{z})\hat{\Psi}(\vec{z}\,'), \tag{1}$$

where $\vec{z} = (x, y, z)$ is the 3D coordinate and $\hat{U}(\vec{z})$ is the effective interaction potential

$$\hat{U}(\vec{z}) = \frac{4\pi a_s}{m}\delta^3(\vec{z})\ . \tag{2}$$

We will always consider elongated gases with $\omega_x \ll \omega_z$ and refer to the $x$-direction as the longitudinal, and the remaining coordinates $\vec{r} \equiv (y, z)$ as the transverse directions. In order to make contact with the experiments of Ref. [14] we use

$$V_\perp(y, t) = \frac{m}{2}\left(\frac{c_1}{c_2}\right)^2\frac{\left(y^2 - c_2^2\left(I^2(t) - I_c^2\right)\right)^2}{I(t) + I_c} + F(t)y, \tag{3}$$

with the values $c_1 = 2\pi \cdot 2.52\,\text{kHz}$, $c_2 = 2.17\,\mu\text{m}$ and $I_c = 0.4$. For $I(t) = I_c$, $V_\perp$ is a quartic potential with a flat bottom and for $I > I_c$, it develops a double well structure. The term $F(t)y$ is used to imprint a phase difference between the gases in the two wells (a precise description of what we mean by this is given below). The explicit form of the functions $I(t)$ and $F(t)$ is given in Sec. 5.1, *cf.* Eqs. (54) and (55).

The idea is to use (1) to describe the splitting of the gas, the phase imprinting and finally the subsequent non-equilibrium dynamics, but to take advantage of the fact that (i) interactions are weak; (ii) the confinement is tight in the y- and z-directions. The combination of these two allows us to project the full three-dimensional theory to a small number of one-dimensional channels

$$H_{\text{proj}}(t) = \sum_{a=0}^{\bar{a}-1}\int dx\ \hat{\psi}_a^\dagger(x, t)\left[-\frac{1}{2m}\frac{\partial^2}{\partial x^2} + \frac{m\omega^2}{2}x^2 + \epsilon_a(t)\right]\hat{\psi}_a(x, t)$$
$$+ \int dx \sum_{a,b,c,d=0}^{\bar{a}-1}\Gamma_{abcd}(t)\,\hat{\psi}_a^\dagger(x, t)\hat{\psi}_b^\dagger(x, t)\hat{\psi}_c(x, t)\hat{\psi}_d(x, t). \tag{4}$$

The resulting Hamiltonian is time-dependent but retardation effects are negligible. Some comments on the projection procedure are provided in Appendix A. We stress that as a result of working with the instantaneous basis of single-particle eigenstates of $-\partial_y^2/2m + V_\perp(y, t)$ our effective one-dimensional field operators $\hat{\psi}_a^\dagger(x, t)$ have an explicit time dependence. After working out how to obtain $H_{\text{proj}}(t)$ from (2) we proceed as follows:

1. We treat the interactions in a time-dependent self-consistent Hartree–Fock approximation (SCHFA). As we are dealing with an effective one-dimensional system we do not allow for the formation of long-range order. The resulting approximation is quite different from Gross-Pitaevskii theory (see e.g. [71–73]). The main attraction of the SCHFA is that it can be implemented straightforwardly, while its main limitation is that it treats interaction effects in a rather crude way. However, it is nonetheless expected to provide a good description as long as the interaction strength is sufficiently weak and the energy density in the system is not too low. We first identify a corresponding parameter regime and then model the Josephson oscillations experiments in this regime.

2. We start with a confining potential that forms a single elongated well and initialize our system in a thermal low-temperature state.

3. We evolve the state under a time-dependent transverse potential $V_\perp(y, t)$ that models the splitting and phase imprinting protocols used in the experiments. This provides us with a characterization of the "initial state" used in the Josephson-like oscillation experiments.

4. Finally we consider the non-equilibrium evolution of the split, phase-imprinted state. We observe damped Josephson-like oscillations.

This paper is organized as follows. In Sec. 2, we describe the low-energy projection used to arrive at the Hamiltonian (4), and the rationale for considering one-dimensional field operators carrying explicit time-dependence. In Sec. 3, we introduce the observables relevant to experiment, and connect them to the Green's functions of one-dimensional field operators that are computed in this paper. Sec. 4 introduces the self-consistent time-dependent Hartree–Fock approximation, and the resulting nonlinear partial differential equations that govern the time-evolution of the experimentally relevant Green's functions. Sec. 5 describes how the initial state of the system is modelled by preparing a gas in a thermal state of a single well and splitting it by a deformation of the trapping potential. The time-dependent definition of the one-dimensional field operators is shown to be an important tool in enabling this model for the preparation sequence. In Sec. 6, numerical results are presented for the time-evolution of experimentally relevant observables after the preparation stage. Density-phase oscillations are observed to be strongly damped over timescales that are comparable to those seen in the experiment.

## 2 Time-dependent projection to one-dimensional channels

We start from the 3D Hamiltonian (1) with $\delta$-interactions (2),

$$H_{3D} = \int d^3\vec{z} \ \hat{\Psi}^\dagger(\vec{z}) \left[ \hat{D}_x + \hat{D}_y(t) + \hat{D}_z + \frac{2\pi a_s}{m} \hat{\Psi}^\dagger(\vec{z})\hat{\Psi}(\vec{z}) \right] \hat{\Psi}(\vec{z}), \qquad (5)$$

where we have defined $\hat{D}_u = -\partial_u^2/2m + m\omega_u^2 u^2/2$ for $u = x, z$, and

$$\hat{D}_y(t) = -\frac{1}{2m} \frac{\partial^2}{\partial y^2} + V_\perp(y, t) . \qquad (6)$$

The 3D Bose field $\hat{\Psi}(\vec{z})$ satisfies the usual bosonic commutation relations. We use a double well potential of the form (3) for $V_\perp(y, t)$ throughout this paper, where the phase imprinting is implemented by the imbalance potential $F(t)y$. To arrive at an effective 1D model, we expand

the 3D field operator in an instantaneous basis of single-particle eigenstates of the quadratic part of the Hamiltonian

$$\hat{\Psi}(\vec{z}\,) = \sum_{a,b,c=0}^{\infty} \chi_a(x)\Phi_b(y,t)\Xi_c(z)\hat{b}_{a,b,c}(t) = \sum_{b,c=0}^{\infty} \Phi_b(y,t)\Xi_c(z)\hat{\tilde{\psi}}_{b,c}(x,t) \qquad (7)$$

Here the single-particle eigenstates fulfil

$$\hat{D}_x \chi_a(x) = \omega_x\big(a + \tfrac{1}{2}\big)\chi_a(x)\,,$$
$$\hat{D}_y(t)\Phi_b(y,t) = \epsilon_b(t)\Phi_b(y,t)\,,$$
$$\hat{D}_z \Xi_c(z) = \omega_z\big(c + \tfrac{1}{2}\big)\Xi_c(z)\,, \qquad (8)$$

and we have defined canonical 1D Bose field operators by

$$\hat{\tilde{\psi}}_{b,c}(x,t) = \sum_{a=0}^{\infty} \chi_a(x)\hat{b}_{a,b,c}(t)\,, \quad [\hat{\tilde{\psi}}_{b,c}(x,t),\hat{\tilde{\psi}}_{b',c'}^{\dagger}(x',t)] = \delta_{b,b'}\delta_{c,c'}\delta(x-x')\,. \qquad (9)$$

At this stage we are dealing with an infinite number of Bose fields. We now exploit the fact that the single-particle eigenvalues $\omega_z(c + 1/2)$ and $\epsilon_b(t)$ constitute very large energy scales for $c \geq 1$ and $b \geq \bar{a}$, and that interactions are weak. This allows us to truncate the expansion of the 3D Bose fields (7) to a small, finite number of channels. The rationale for working with explicitly time-dependent single-particle states rather than working in a fixed basis is that a subset chosen to span the low-energy subspace of the free part of the Hamiltonian (5) at $t = 0$ will in general only span the low-energy subspace at later times if we include a large number of channels. This would make the truncation much less efficient. Let us now give the details of the truncation procedure outlined above. If $\omega_z \gg \omega_x$, we expect the dynamics to be frozen into the lowest single-particle eigenstates in the $z$-direction. We can then project to the corresponding low-energy subspace by truncating the expansion (7) to the $c = 0$ term. In the $y$-direction, the double well $V_\perp(y,t)$ gives rise to more states in the low-energy sector than just the ground state. We therefore need to retain multiple single-particle eigenstates $\Phi_b(y,t)$. These wave functions are explicitly time-dependent eigenstates of the double well operator $\hat{D}_y(t)$ from Eq. (6), with eigenvalues $\epsilon_a(t)$. If these eigenvalues show a gap above energy $\epsilon_{\bar{a}-1}(t)$ that is large compared to all other energy scales in the problem for all times, the expansion (7) can be truncated at $a = \bar{a}$. The resulting projection of the 3D field operator to the low-energy sector then reads

$$\hat{\Psi}(\vec{z}\,) \approx \Xi_0(z)\sum_{a=0}^{\bar{a}-1} \Phi_a(y,t)\hat{\psi}_a(x,t)\,, \qquad (10)$$

where we have defined

$$\hat{\psi}_a(x,t) \equiv \hat{\tilde{\psi}}_{a,0}(x,t)\,, \qquad (11)$$

which satisfies $[\hat{\psi}_a(x,t),\hat{\psi}_b^{\dagger}(x',t)] = \delta_{ab}\delta(x-x')$ for all times. When starting from the 3D Hamiltonian (5), inserting the projected operator (10) and integrating in the $y,z$-directions leads to a model for $\bar{a}$ species of bosons,

$$H_{1D}^{(\bar{a})}(t) = \sum_{a=0}^{\bar{a}-1} \int dx\, \hat{\psi}_a^{\dagger}(x,t)\left[-\frac{1}{2m}\frac{\partial^2}{\partial x^2} + \frac{m\omega^2}{2}x^2 + \epsilon_a(t)\right]\hat{\psi}_a(x,t)$$

$$+ \int dx \sum_{a,b,c,d=0}^{\bar{a}-1} \Gamma_{abcd}(t)\hat{\psi}_a^{\dagger}(x,t)\hat{\psi}_b^{\dagger}(x,t)\hat{\psi}_c(x,t)\hat{\psi}_d(x,t)\,, \qquad (12)$$

with coupling constants that are given by overlap tensors

$$\Gamma_{abcd}(t) = a_s \sqrt{\frac{2\pi\omega_z}{m}} \int dy \; \Phi_a^*(y,t)\Phi_b^*(y,t)\Phi_c(y,t)\Phi_d(y,t) \, . \tag{13}$$

Corrections to (12) will be negligible as long as the following conditions hold:

- Interactions are small. This holds by construction.

- The initial occupation numbers $\mathrm{Tr}[\rho(0) \; b_{a,b,c}^\dagger(0)b_{a,b,c}(0)]$, where $\rho(0)$ is the density matrix at time $t = 0$, are very small for $c > 0$ and $b \geq \bar{a}$. We ensure that this is the case by working with an initial thermal density matrix at a sufficiently low energy density compared to $\epsilon_{\bar{a}}(t)$. Experimentally this condition could be fulfilled by making $V_\perp(y,t)$ sufficiently tight.

- The transverse potential is changed slowly enough so that $\mathrm{Tr}[\rho(t) \; b_{a,b,c}^\dagger(t)b_{a,b,c}(t)]$ remain very small for $c > 0$ and $b \geq \bar{a}$. This provides a (rather obvious) restriction on the experimental protocol.

In equilibrium it is straightforward to evaluate the corrections to (12) and we outline the necessary steps in Appendix A. Perturbatively integrating out the high-energy channels above some cutoff generates all two, four and six boson interactions between the low-energy channels allowed by particle conservation. The interactions are very slightly retarded and non-local in space (the corresponding scales are set by the cut-off energy) but are negligible compared to the terms retained in (12). An analogous analysis can be in principle be carried out in the time-dependent situation of interest here, but as we don't require the corrections we do not follow this line of enquiry further.

## 2.1 Connection to previous literature

For time-independent double-well potentials with a very high tunnel-barrier, the lowest two single-particle eigenstates $\Phi_{0,1}(y)$ are approximately given by symmetric and anti-symmetric combinations of wave packets $g_{L,R}(y)$ that are localized in the left and right wells

$$\Phi_{0,1}(y) = (g_R(y) \pm g_L(y))/\sqrt{2} \, . \tag{14}$$

We can then define left- and right-localized one-dimensional Bose operators $\hat{\psi}_{L,R} \approx (\hat{\psi}_0 \pm \hat{\psi}_1) \times 1/\sqrt{2}$. Inserting these definitions into Eq. (4) with $\bar{a} = 2$ leads to the model

$$H_{1D} \to H_{LL}\big[\hat{\psi}_L\big] + H_{LL}\big[\hat{\psi}_R\big] - \frac{\epsilon_1 - \epsilon_0}{2} \int_0^L dx \, \big(\hat{\psi}_L^\dagger(x)\hat{\psi}_R(x) + \text{h.c.}\big) \, , \tag{15}$$

of two Lieb-Liniger Hamiltonians

$$H_{LL}\big[\hat{\psi}\big] = \int dx \, \hat{\psi}^\dagger(x)\left[-\frac{1}{2m}\frac{\partial^2}{\partial x^2} + \frac{m\omega^2}{2}x^2\right]\hat{\psi}(x) + \mathfrak{g} \int dx \, \hat{\psi}^\dagger(x)\hat{\psi}^\dagger(x)\hat{\psi}(x)\hat{\psi}(x) \, , \tag{16}$$

connected by a tunnel-coupling term. The coupling constant of the Lieb-Liniger model is given by

$$\mathfrak{g} = \int dx \, g_L^*(x)g_L^*(x)g_L(x)g_L(x) = \int dx \, g_R^*(x)g_R^*(x)g_R(x)g_R(x) \, . \tag{17}$$

All other overlap tensors involving four combinations of $g_{L,R}(x)$ vanish if the two wells are separated by a high tunnel barrier, so that the only coupling between the left and right gases is given by the tunneling term proportional to $(\epsilon_1 - \epsilon_0)/2$. Eq. (15) is the Hamiltonian that is studied in most of the literature, following Ref. [45]. In this paper, we will instead focus on the more general Hamiltonian (12).

## 2.2 Three channel model

So far we have kept the number $\bar{a}$ of one-dimensional channels in our theory arbitrary. In practice it turns out to be sufficient to work with $\bar{a} = 3$ in order to accommodate the experimental situation realized in the Vienna group. The trapping geometries in these experiments are chosen so as to strongly suppress the occupation of the second excited level ($a = 2$). By including this suppressed level in our simulations and staying close to the experimental energy scales and trapping frequencies we can therefore ensure that we are in a regime where the occupation of the (time-dependent) third single-particle excited level can be safely neglected. The energy scales relevant to this reasoning are displayed in Fig. 1. This figure shows that at $t = 0$, the single-particle energy of the third level, which is neglected in our simulations, differs from the single-particle ground state energy by $\epsilon_3(0) - \epsilon_0(0) \approx 2.5\,k_B T$. This means that it is reasonable to neglect the occupations of the third and higher levels at $t = 0$. For later times, there are two requirements to be able to keep neglecting these levels. Firstly, we rely on the interactions being weak, and secondly, we need the change in $V_\perp(y, t)$ to be slow enough.

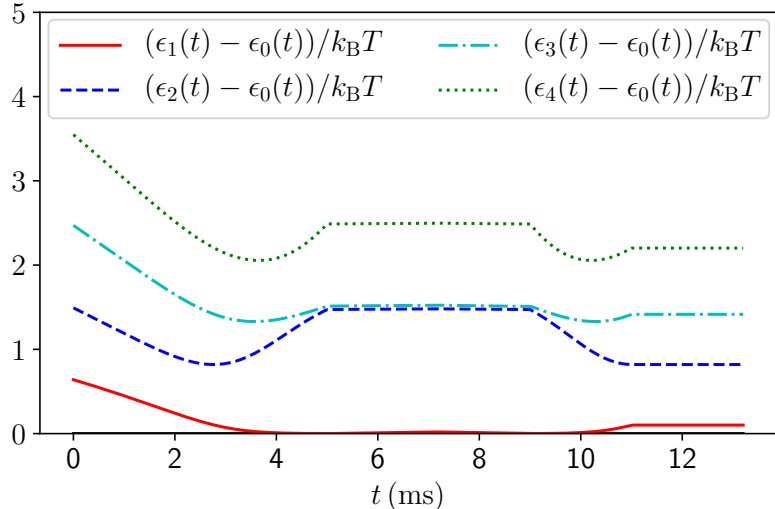

Figure 1: Time-dependent energies $\epsilon_j(t)$ of the lowest eigenstates of the transverse single-particle Hamiltonian (6), using the double well potential defined in and below Eq. (3). The energies are displayed via their difference with $\epsilon_0(t)$, in units of $k_B T$, with $T = 60\,\text{nK}$.

## 3 Measurements and Green's functions

### 3.1 Measured operator in time of flight

The Bose gases in the double well can be probed through matter-wave interferometry [31, 74, 75]. After a tunable time $t_0$ spent in the double well, the Bose gases are released by turning off the trapping potential. This causes them to expand and overlap in three-dimensional space, and eventually their combined density is measured by absorption imagining after a "time of flight" $t_1$. The theoretical description of this measurement process in the framework of a low-energy theory description is described in detail in Ref. [38] in the simpler case when two transverse single-particle states, given by Gaussian wave packets in the left and right wells respectively, are kept in the projection to an effectively one-dimensional model. We will briefly recapitulate this construction, before expanding it to the case of more than two levels

that are not perfectly localized in the wells. The absorption imagining can be thought of as a von-Neumann measurement of the boson density at time $t_0 + t_1$

$$\hat{\rho}_{\text{tof}}(\mathbf{r}) = \hat{\Psi}^{\dagger}(\mathbf{r})\hat{\Psi}(\mathbf{r}) . \tag{18}$$

The density operator is diagonal in the position eigenbasis $\{|\mathbf{r}_1, \ldots, \mathbf{r}_N\rangle\}$ which implies that the measurement outcomes are particle positions $\sum_{j=1}^{N} \delta(\mathbf{r} - \mathbf{r}_j)$ and the associated probability distribution is

$$P(\mathbf{r}_1, \ldots, \mathbf{r}_N; t_0 + t_1) = \langle \mathbf{r}_1, \ldots, \mathbf{r}_N | \varrho(t_0 + t_1) | \mathbf{r}_1, \ldots, \mathbf{r}_N \rangle , \tag{19}$$

where $\varrho(t_0 + t_1)$ is the density matrix of the system at time $t_0 + t_1$. The moments of this probability distribution are

$$M_n(\mathbf{r}_1, \ldots, \mathbf{r}_n) = \text{Tr}\left[ \varrho(t_0 + t_1) \, \hat{\mathcal{O}}^{\dagger}(\mathbf{r}_1, \ldots, \mathbf{r}_n)\hat{\mathcal{O}}(\mathbf{r}_1, \ldots, \mathbf{r}_n) \right],$$
$$\hat{\mathcal{O}}(\mathbf{r}_1, \ldots, \mathbf{r}_n) = \hat{\Psi}(\mathbf{r}_1) \ldots \hat{\Psi}(\mathbf{r}_n) . \tag{20}$$

The density matrix at time $t$ is given by

$$\varrho(t) = \mathcal{U}(t, 0)\varrho(0)\mathcal{U}^{\dagger}(t, 0) , \quad \mathcal{U}(t, t_0) = T \exp\left( -i \int_{t_0}^{t} dt' H_{3\text{D}}(t') \right) . \tag{21}$$

In the Heisenberg picture we have

$$M_n(\mathbf{r}_1, \ldots, \mathbf{r}_N) = \text{Tr}\left[ \varrho(0) \left( \hat{\mathcal{O}}^{(H)}(\mathbf{r}_1, \ldots, \mathbf{r}_n, t_0 + t_1) \right)^{\dagger} \hat{\mathcal{O}}^{(H)}(\mathbf{r}_1, \ldots, \mathbf{r}_n, t_0 + t_1) \right], \tag{22}$$

where the Heisenberg-picture field operators are given by

$$\hat{\Psi}^{(H)}(\mathbf{r}, t) = \mathcal{U}^{\dagger}(t, 0)\hat{\Psi}^{(H)}(\mathbf{r}, 0)\mathcal{U}(t, 0) . \tag{23}$$

The approach of Ref. [75] is to relate the quantum state of the system after time-of-flight to the state at the time of trap release by assuming that interactions are negligible during the time of flight. This is a reasonable assumption since the gas, which is no longer constrained and expands in 3D, very quickly becomes highly dilute. The free, transverse expansion is then effectuated by the evolution operator

$$\mathcal{U}(t_1 + t_0, t_0) \approx e^{-it_1\left( \hat{P}_x^2 + \hat{P}_y^2 + \hat{P}_z^2 \right)/2m} . \tag{24}$$

This allows us to relate the Heisenberg picture field operators at times $t_0 + t_1$ and $t_0$

$$\hat{\Psi}^{(H)}(\mathbf{r}, t_0 + t_1) = \int d^3\mathbf{r}' \, G_0(\mathbf{r} - \mathbf{r}', t_1) \, \hat{\Psi}^{(H)}(\mathbf{r}', t_0) , \tag{25}$$

where $G_0(\mathbf{r}, t)$ is the propagator of the non-interacting boson Hamiltonian describing the free expansion. Now we exploit the fact that the initial density matrix $\rho(0)$ involves only the low-energy sector, i.e. states in which only very few transverse modes are occupied. This allows us to project the field operators $\hat{\Psi}^{(H)}(\mathbf{r}, t_0)$ and concomitantly the operators $\hat{\mathcal{O}}^{(H)}(\mathbf{r}_1, \ldots, \mathbf{r}_n, t_0 + t_1)$ in the expression (22) for the moments $M_n$ to the low-energy description

$$\hat{\Psi}^{(H)}(\mathbf{r}, t_0) \approx \Xi_0(z) \sum_{a \in S} g_a(y, t) \hat{\psi}_a^{(H)}(x, t) , \tag{26}$$

where $S$ is some set of indices labeling single-particle states $g_a(y, t)$ in the transverse direction. The equations of motion of the Heisenberg picture operators $\hat{\psi}_a^{(H)}(x, t)$ are derived below in

section 4. In [38], this set $S = \{L, R\}$ refers to single-particle states with no explicit time-dependence that are localized in the left and right wells, respectively. In the following we will focus on the average over many absorption images

$$M_1(\mathbf{r}) = \text{Tr}\Big[\varrho(0)\, \hat{\rho}_{\text{tof}}^{(H)}(\mathbf{r}, t_0 + t_1)\Big].$$ (27)

Carrying out the convolutions we obtain

$$\hat{\rho}_{\text{tof}}^{(H)}(\mathbf{r}, t_0 + t_1) \approx |\bar{\Xi}_0(z, t_1)|^2 \sum_{i,j \in S} A_{ij}(y, t_0, t_1) \int dx' d\tilde{x}\, G_0^*(x - x', t_1) G_0(x - \tilde{x}, t_1)$$
$$\times \Big(\hat{\psi}_i^{(H)}(x', t_0)\Big)^\dagger \hat{\psi}_j^{(H)}(\tilde{x}, t_0)\,.$$ (28)

Here we have defined

$$A_{ij}(y, t_0, t_1) = \bar{g}_i^*(y, t_0, t_1)\bar{g}_j(y, t_0, t_1), \quad i, j \in S\,,$$
$$\bar{g}_j(y, t_0, t_1) \equiv \int dy'\, \sqrt{\frac{m}{2\pi i t_1}} \exp\Big(i\frac{m}{2t_1}(y - y')^2\Big) g_j(y', t_0)\,,$$ (29)

and an analogous expression is obtained for $\bar{\Xi}_0(z)$. The higher moments $M_{n>1}$ can be related to expectation values in the low-energy description in the same way.

In many works [44,75] it is assumed that the longitudinal expansion has little effect (even though it can be straightforwardly taken into account in a low-energy field theory framework in some cases [38]). This assumption is based on the state at the time of trap release: since the gas is spatially very constrained in the transverse directions, its momentum distribution in these directions is much broader than in the longitudinal direction. As a result, the time scale for expansion in the longitudinal direction far exceeds that for the transverse directions. If the time of flight $t_1$ is short, this suggests the approximation of neglecting longitudinal expansion altogether, replacing the free evolution operator (24) by

$$\mathcal{U}(t_1 + t_0; t_0) \approx e^{-it_1\left(\hat{P}_y^2 + \hat{P}_z^2\right)/2m}\,.$$ (30)

This results in a simplified expression for the operator $\hat{\rho}_{\text{tof}}^{(H)}$

$$\hat{\rho}_{\text{tof}}^{(H)}(\mathbf{r}, t_1, t_0) \approx |\bar{\Xi}_0(z, t_1)|^2 \sum_{i,j \in S} A_{ij}(y, t_0, t_1)\Big(\hat{\psi}_i^{(H)}(x, t_0)\Big)^\dagger \hat{\psi}_j^{(H)}(x, t_0)\,.$$ (31)

We will use this approximate expression in much of the remainder of this work.

## 3.2 Green's functions of interest

In what follows, we will derive equations of motion for the Green's functions of the 1D Bose fields, defined as

$$C_{ij}(x, x', t) \equiv \langle \hat{\psi}_i^\dagger(x, t)\hat{\psi}_j(x', t)\rangle\,.$$ (32)

Solving these numerically gives us access to the expectation value of $\hat{\rho}_{\text{tof}}^{(H)}(\mathbf{r}, t_0 + t_1)$ as well as averages of Fourier-transformed quantities like (41). In order to connect to the experimental works [10, 14, 15] we have to account for the fact that the data extracted from absorption imaging has been analyzed in terms of the number/phase representation for an effective

two-channel model. Denoting the corresponding Bose field operators by $\hat{\psi}_{L,R}$ we can define averages of the relative density and phase via

$$
\begin{aligned}
\varphi(x,t) &= \mathrm{Arg}\,\langle \hat{\psi}_L^\dagger(x,t)\hat{\psi}_R(x,t)\rangle, \\
n(x,t) &= \langle \hat{\psi}_L^\dagger(x,t)\hat{\psi}_L(x,t)\rangle - \langle \hat{\psi}_R^\dagger(x,t)\hat{\psi}_R(x,t)\rangle.
\end{aligned}
\tag{33}
$$

In order to connect to these quantities we need to express $\hat{\psi}_{L,R}$ in terms of operators in our three-channel model. As interactions are weak this transformation can be taken to be linear. To be specific let us work with an effective three-channel model, i.e. $\bar{a}=3$. We then carry out a change of basis such that

$$
\hat{\psi}_\alpha(x,t) = \sum_{j=0}^{2} c_j^{(\alpha)}(t)\hat{\psi}_j(x,t), \quad \alpha = L,R,e.
\tag{34}
$$

We have introduced a third, "excited" boson species $\hat{\psi}_e$ to be able to span the full space of 3 transverse levels. The set of labels used in Eq. (26) thus becomes $S = \{L,R,e\}$, so that Eq. (26) is equivalent to Eq. (10) under the identifications

$$
\Phi_j(y,t) = \sum_{\alpha = L,R,e} c_j^{(\alpha)}(t) g_\alpha(y,t), \quad j = 0,1,2.
\tag{35}
$$

The transformation matrices $c_j^{(\alpha)}(t)$ are chosen with orthonormal rows and columns, so that they translate between the basis of single-particle eigenstates $\Phi_{0,1,2}(y,t)$ of the transverse operator $\hat{D}_y(t)$ and another basis that contains left- and right-localized wave functions $g_{L,R}(y,t)$ as well es a third wave function, $g_e(y,t)$.

In [38], the wave functions $g_{L,R}(y,t)$ were simply given by (anti)symmetric combinations of $\Phi_0$ and $\Phi_1$. However, the presence of the third wave function $g_e(y,t)$ now creates ambiguity, meaning that the $c_j^{(\alpha)}(t)$ can be defined in multiple ways. We will give two options here.

**Choice 1:** Following Ref. [38], we simply choose

$$
\begin{pmatrix} c_0^{(L)} & c_0^{(R)} & c_0^{(e)} \\ c_1^{(L)} & c_1^{(R)} & c_1^{(e)} \\ c_2^{(L)} & c_2^{(R)} & c_2^{(e)} \end{pmatrix}(t) = \frac{1}{\sqrt{2}} \begin{pmatrix} 1 & 1 & 0 \\ 1 & -1 & 0 \\ 0 & 0 & \sqrt{2} \end{pmatrix} \quad \forall\, t\,.
\tag{36}
$$

**Choice 2:** Since the double well is centered around $y = 0$, we find the vector $c_j^{(L)}(t)$ by minimizing $\int_0^\infty dy\,|g_L(y,t)|^2$ subject to the constraint $\sum_j |c_j^{(L)}(t)|^2 = 1$. This fixes $g_L(y,t)$ as the single-particle wave function in the space spanned by $\Phi_{0,1,2}(y,t)$ with the smallest possible probability for the particle to be found at $y > 0$, i.e. in the right well. *Mutatis mutandis* for $c_j^{(R)}(t)$. The third vector $c_j^{(e)}(t)$ is then defined as the orthogonal complement of the vectors $c_j^{(L)}(t)$ and $c_j^{(R)}(t)$. The experimentally relevant parameters for the double well potential are given below Eq. (3), with $0.5 \le I \le 0.6$ for the oscillation stage. For most of these values choices 1 and 2 lead to very similar values of $c_j^{(j)}(t)$ and for $I \ge 0.55$, the values are practically indistinguishable. We will therefore present results for the much simpler Choice 1, and comment on the changes that occur for Choice 2 wherever they are relevant.

Using Choice 1 and Eq. (33), the average relative density and phase (33) are given by

$$
\begin{aligned}
\varphi(x,t) &\equiv \mathrm{Arg}\, C_{LR}(x,x,t) \\
&= \mathrm{Arg}\,\frac{1}{2}\left[ C_{00}(x,x,t) - C_{01}(x,x,t) + C_{01}^*(x,x,t) - C_{11}(x,x,t) \right],
\end{aligned}
\tag{37}
$$

$$
n(x,t) \equiv \langle \hat{n}(x,t)\rangle = C_{LL}(x,x,t) - C_{RR}(x,x,t) = 2\mathrm{Re}\, C_{01}(x,x)\,.
\tag{38}
$$

Another quantity of experimental interest is the *mean interference contrast*, which we define as

$$\mathcal{C}(x,t) = \frac{2\,|C_{LR}(x,x,t)|}{|C_{LL}(x,x,t) + C_{RR}(x,x,t)|}. \tag{39}$$

### 3.3 Experimental data analysis and its relation to Green's functions

As discussed above the average over many absorption images gives access to

$$\langle \hat{\rho}_{\text{tof}}^{(H)}(\mathbf{r}, t_1 + t_0)\rangle \approx |\bar{\Xi}_0(z, t_1)|^2 \sum_{i,j \in \{L,R,e\}} A_{ij}(y, t_0, t_1)\left\langle \left(\hat{\psi}_i^{(H)}(x, t_0)\right)^{\dagger} \hat{\psi}_j^{(H)}(x, t_0)\right\rangle \tag{40}$$

in the three-channel model. We refer to Eq. (28) for the case when longitudinal expansion is taken into account. We will now show that the phase $\varphi(x, t_0)$ of interest in Eq. (37) can be extracted from $M_1(\mathbf{r})$ by taking a suitable partial Fourier transform

$$\mathcal{F}_q\left[\langle \hat{\rho}_{\text{tof}}^{(H)}(\mathbf{r}, t_1 + t_0)\rangle\right] = \int dy\, e^{-iqy}\langle \hat{\rho}_{\text{tof}}^{(H)}(\mathbf{r}, t_1 + t_0)\rangle. \tag{41}$$

In the simpler case of gases whose transverse single-particle states are given by Gaussian wave packets in the left and right wells respectively the choice $q = md/t_1$, where $d$ is the distance between the wells' minima, gives access to the relative phase, see *e.g.* Ref. [38].

It is important to check how this situation is affected by the presence of the third channel and by the fact that the localized single-particle states are not given by perfect Gaussian wave packets. Studying the amplitudes $A_{ij}$ numerically for a given double well potential, we can establish which terms in (40) contribute at the wave vector $q = md/t_1$ for a realistic potential in the three channel model. As shown in Fig. 2, $A_{LR}$ has a marked peak in Fourier space around $q = md/t_1$. The diagonal terms $\sim A_{ii}$ only contribute around $q \approx 0$. The terms $\sim A_{Le}$ and $\sim A_{Re}$ do contribute at higher wave vectors, but their Fourier transforms both become very small around $|q| = md/t_1$ for all values of the double well (3) we consider. Moreover, the occupation of the "excited" transverse wave function $g_e(y, t)$ is much smaller than that of the wave functions $g_{L,R}(y, t)$. For these reasons, the Fourier transform (41) at $q = md/t_1$ is well approximated by

$$\mathcal{F}_q\left[\langle \hat{\rho}_{\text{tof}}^{(H)}(\mathbf{r}, t_1 + t_0)\rangle\right]\bigg|_{q=md/t_1} \propto C_{LR}(x, x, t_0), \tag{42}$$

whose argument then provides the relative phase of interest

$$\varphi(x, t_0) = \text{Arg}\,\mathcal{F}_q^{(y)}\left[\left\langle \hat{\rho}_{\text{tof}}^{(H)}(\mathbf{r}, t_1, t_0)\right\rangle\right]\bigg|_{q=md/t_1} \approx \text{Arg}\,C_{LR}(x, x, t_0). \tag{43}$$

The above holds for expectation values with respect to states belonging to the low-energy subspace. If on the other hand one works with a trapping potential where $A_{Le}$ and $A_{Re}$ do not have small Fourier components at $q = md/t_1$ and if the occupation of $g_e(y, t)$ is not small, the identification (42) might fail. Another likely scenario is that the value of $|q| = md/t_1$ cannot be established to sufficient precision. In such cases, Eq. (40) shows how different boson bilinears contribute to the measured density after time-of-flight, using numerical evaluations of the amplitudes $A_{ij}(y, t_0, t_1)$.

Our way of extracting the relative phase from the average over many absorption images should be contrasted to the way in which the experiments [10, 14, 15] have been analyzed. In these works a value for a relative phase $\phi(x, t_0)$ is extracted *for each (typical) absorption image* by fitting the observed density profile to an expression of the form

$$\rho_{\text{tof}}(\mathbf{r}, t_1 + t_0) \approx |f(y, z, t_1)|^2\left(1 + \cos\left(\phi(x, t_0) + ydm/t_1\right)\right), \tag{44}$$

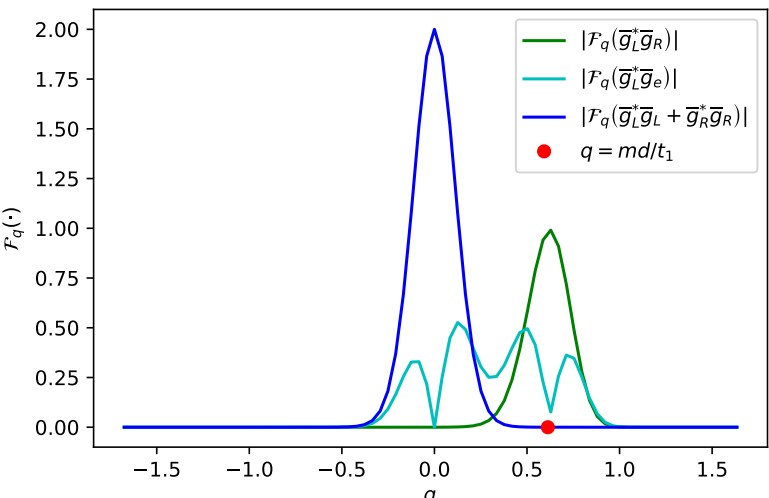

Figure 2: Fourier transformed products of single-particle wave functions after time-of-flight $\overline{g}_{L,R}(y,t_1)$ occurring in Eq. (31), for the parameters given below Eq. 3 with $I = 0.5$. The cross term $\overline{g}_L^*(y,t_1)\overline{g}_R(y,t_1)$ (green) shows a peak around $q = md/t_1$, whereas $\overline{g}_L^*(y,t_1)\overline{g}_e(y,t_1)$ (cyan) becomes small there. The same can be said about the other cross terms involving $\overline{g}_e$. This allows to extract $\varphi_a(x)$ using Eq. (43).

where $f(y,z,t_1)$ is a Gaussian envelope. The data is then analyzed in terms of the average $\overline{\phi(x,t_0)}$ over many shots. An interesting open question is to establish the precise relation between $\overline{\phi(x,t_0)}$ and $\varphi(x,t_0)$ extracted from the average over many images.

## 4 Hartree–Fock time evolution

Having established how Green's functions are related to averages over experimental measurements, we now consider their time evolution. We do so in the Heisenberg picture, indicated with a superscript $(H)$, and consider the equations of motion for the 1D field operators,

$$i\frac{d}{dt}\hat{\psi}_a^{(H)}(x,t) = \left[\hat{\psi}_a^{(H)}(x,t), H_{1D}^{(\overline{a},H)}(t)\right] + iU^\dagger(t)\frac{\partial}{\partial t}\hat{\psi}_a(x,t)U(t). \tag{45}$$

Here $U(t)$ is the time-evolution operator

$$U(t) = T\exp\left(-i\int_0^t dt' H_{1D}^{(\overline{a})}(t')\right), \tag{46}$$

and the additional, explicit time-derivative is nonzero due to the time-dependent definition of $\hat{\psi}_a(x,t)$, via the corresponding eigenstates $\Phi_a(y,t)$ of the transverse potential $V_\perp(y,t)$. In order to work out the last term on the right hand side of (45) we revert to the expansion of the Bose field into channels without projection (7)

$$0 = \frac{\partial \hat{\Psi}(\vec{x})}{\partial t} = \frac{\partial}{\partial t}\sum_{b,c}\Phi_b(y,t)\Xi_c(z)\hat{\tilde{\psi}}_{b,c}(x,t). \tag{47}$$

The orthonormality of the single-particle wave functions then implies that

$$\frac{\partial}{\partial t}\hat{\tilde{\psi}}_{bc}(x,t) = \sum_{d=0}^\infty B_{cd}^*(t)\hat{\tilde{\psi}}_{cd}(x,t), \quad B_{ab}(t) = -\int dy\,\Phi_a(y,t)\dot{\Phi}_b^*(y,t). \tag{48}$$

Using our assumption that the transverse potential is varying sufficiently slowly in time we can project these equations to our model with $\bar{a}$ transverse channels

$$U^{\dagger}(t)\frac{\partial}{\partial t}\hat{\psi}_a(x,t)U(t) \approx \sum_{b=0}^{\bar{a}-1} B_{ab}^*(t)\hat{\psi}_b^{(H)}(x,t). \tag{49}$$

Physically, this term in the equation of motion (45) keeps track of transitions $a \to b$ to different levels due to time-dependence in $V_{\perp}(y,t)$. In what follows, we will drop the superscript $(H)$ and fix $\bar{a} = 3$.

We now make the Hartree–Fock approximation for the interaction term,

$$\begin{aligned}
\hat{\psi}_a^{\dagger}(x,t)\hat{\psi}_b^{\dagger}(x,t)\hat{\psi}_c(x,t)\hat{\psi}_d(x,t) \to \\
C_{ac}(x,x,t)\hat{\psi}_b^{\dagger}(x,t)\hat{\psi}_d(x,t) + C_{bd}(x,x,t)\hat{\psi}_a^{\dagger}(x,t)\hat{\psi}_c(x,t) \\
+C_{ad}(x,x,t)\hat{\psi}_b^{\dagger}(x,t)\hat{\psi}_c(x,t) + C_{bc}(x,x,t)\hat{\psi}_a^{\dagger}(x,t)\hat{\psi}_d(x,t)\,.
\end{aligned} \tag{50}$$

Using the symmetry of $\Gamma_{abcd}(t)$ the truncated Heisenberg equation (45) then yields the self-consistent equations

$$\begin{aligned}
\frac{d}{dt}C_{ab}(x,x',t) = i\left(\hat{D}_x - \hat{D}_{x'} + \epsilon_a(t) - \epsilon_b(t)\right)C_{ab}(x,x',t) \\
+ 4iG_{ac}^*(x,t)C_{cb}(x,x',t) - 4iG_{bc}(x',t)C_{ac}(x,x',t)\,,
\end{aligned} \tag{51}$$

describing the time evolution of the Green's functions of interest, namely $C_{ab}(x,x',t)$ with $a,b = 0,\ldots,\bar{a}-1$. The HF approximation is equivalent to neglecting all higher connected $n$-point functions other than these Green's functions. The self-consistency of the HF scheme is implemented by the effective potentials

$$G_{bc}(x,t) = \sum_{a,d=0}^{2} \Gamma_{abcd}(t)\,C_{ad}(x,x,t) + \frac{i}{4}B_{bc}^*(t)\,, \tag{52}$$

with $B(t)$ given by Eq. (48).

The system of Eqs. (51) can be solved numerically. In our implementation, we use a mixed implicit-explicit method for the propagation in time, employing a Crank–Nicholson scheme for the terms linear in Green's functions and a first order forward Euler method for the nonlinear terms. We work on a 2D square spatial grid of linear size $250\,\mu$m, using $1000\times1000$ grid points and approximating spatial derivatives by fourth order finite differences. We have checked convergence with respect to the grid spacing as well as the time step, which is $0.015$ ms in the figures presented below. At each time step during the preparation sequence, the matrix $B(t)$ given by Eq. (49) is computed for the lowest $\bar{a}$ eigenfunctions corresponding to the potential $V_{\perp}(y,t)$.

## 4.1 Quality of the SCHF approximation in equilibrium

An important question is how well we expect the HF approximation to work. It is well known [79] that at sufficiently low temperatures, 1D Bose gases form a quasi-condensate which is not well captured in the HF approximation. Specifically, the 1D boson density develops a central density peak which is underestimated by HF calculations. To make this precise we consider the simpler case of the Lieb–Liniger model in a harmonic trap $V_{\parallel}(x)$, where we can compare finite-temperature HF computations to results using Yang–Yang thermodynamics combined with the Local Density Approximation (YY+LDA). The LDA method is expected to provide

highly accurate results in the appropriate parameter regime and its application to the Lieb–Liniger model has been described in detail in [78]. It has been successfully tested in experimental settings [80] and we will use it to compute the quantities

$$\Delta_1 = \int dx \left( \langle \psi^\dagger(x)\psi(x) \rangle_{\text{YY+LDA}} - \langle \psi^\dagger(x)\psi(x) \rangle_{\text{HF}} \right) / N_{\text{HF}} \,, \tag{53}$$

$$\Delta_2 = \int dx \left( \sqrt{\langle (\psi^\dagger(x))^2 (\psi(x))^2 \rangle}_{\text{YY+LDA}} - \sqrt{\langle (\psi^\dagger(x))^2 (\psi(x))^2 \rangle}_{\text{HF}} \right) / N_{\text{HF}} \,,$$

with $N_{\text{HF}} = \int dx \, \langle \psi^\dagger(x)\psi(x) \rangle_{\text{HF}}$. The expectation values $\langle \cdot \rangle_{\text{HF}}$ are computed by the methods of Sec. 5.2 and using Wick's theorem. The expectation values $\langle \cdot \rangle_{\text{YY+LDA}}$, on the other hand, are computed by numerically solving the thermodynamic Bethe Ansatz equations at finite temperature [81], using a chemical potential that is slowly varying in space $\mu(x) = \mu_0 - V_\parallel(x)$. For $\Delta_2$, the Hellman-Feynman theorem must be used in addition [78]. The criterion for LDA to be applicable [78] can be checked *a posteriori*, and is found to be satisfied everywhere away from the boundaries of the gas for our parameters.

A comparison between HF and YY+LDA for density profiles $\rho_0 = \langle \psi^\dagger(x)\psi(x) \rangle$ of a single gas is presented in Fig. 3. We see that while the HF approximation works quite well overall, it does underestimate the central peak. This failure occurs above a certain particle number, and the number where this cross-over occurs decreases with temperature. We will therefore

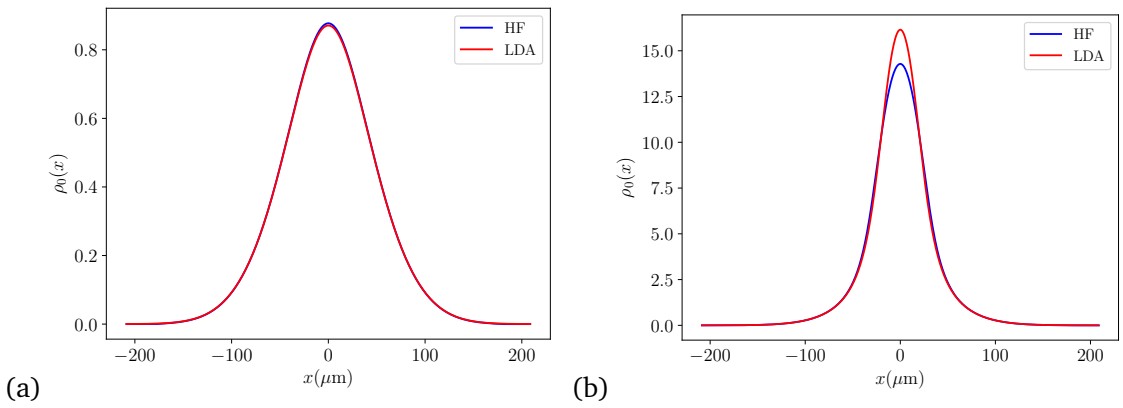

Figure 3: Comparison between density profiles of a single gas in a harmonic longitudinal potential with $\omega_x = 2\pi \cdot 12.5\,\text{Hz}$, computed in Yang–Yang thermodynamics with LDA (red), versus HF (blue), at $T = 60\,\text{nK}$. For a low particle number (panel (a), $N = 99$), the correspondence is good, whereas for $N = 986$ (b), the central density peak is underestimated in HF.

work at a relatively high temperature of $T = 60\,\text{nK}$ in what follows. To make sure our particle number does not exceed the cross-over where HF fails, we have plotted $\Delta_{1,2}$ for a range of particle numbers and longitudinal trapping frequencies in Fig. 4. This allows to monitor the quality of HF in the initial state for the parameters of our simulation. In particular, in the regime where $\Delta_1$ is small the value of $\Delta_2$ provides an indication of the strength of connected 4-point correlations, which vanish in HF. For $T = 60\,\text{nK}$ and $N \lesssim 200$, Fig. 4(b) shows it to be small. We note that our self-consistent Hartree–Fock results can in principle be improved upon for weak interactions and small particle numbers using only the self-consistently determined Green's function $\langle \psi^\dagger(x)\psi(x) \rangle_{\text{HF}}$, combined with perturbation theory. Rather than simply using Wick's theorem to compute the expectation values $\langle \cdot \rangle_{\text{HF}}$ occurring in Eq. (53) in terms of $\langle \psi^\dagger(x)\psi(x) \rangle_{\text{HF}}$, one could include perturbative corrections to these results, working in powers of the interaction strength and performing contractions using the self-consistently determined

Green's function $\langle \psi^\dagger(x)\psi(x)\rangle_{\mathrm{HF}}$. We have checked the first order term and observed that it brings the result closer to the YY+LDA result for small particle numbers ($N \lesssim 300$), whereas the correction starts to diverge for larger particle numbers.

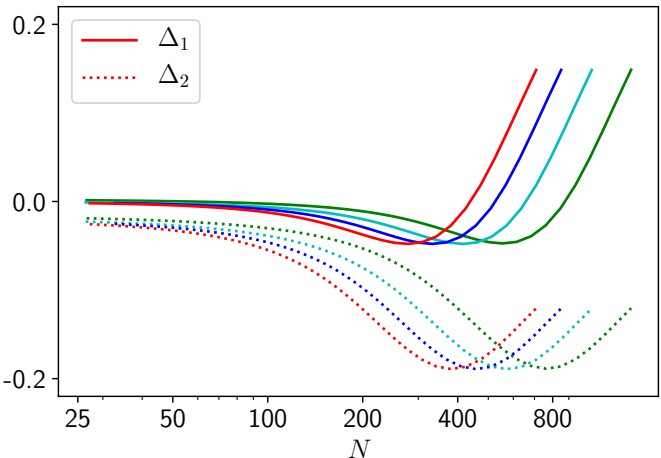

Figure 4: Errors $\Delta_{1,2}$ between HF and YY+LDA from Eq. (53) for $T = 60\,\mathrm{K}$. The colors correspond to $\omega_\parallel = 2\pi \cdot 7.5\,\mathrm{Hz}$ (green), $\omega_\parallel = 2\pi \cdot 10\,\mathrm{Hz}$ (cyan), $\omega_\parallel = 2\pi \cdot 12.5\,\mathrm{Hz}$ (blue) and $\omega_\parallel = 2\pi \cdot 15\,\mathrm{Hz}$ (red).

## 5 Initial state and gas splitting

### 5.1 Preparation sequence

We now have an equation of motion at hand for the relevant Green's functions that enter observables. Starting from an appropriate initial state, we can thus simulate the effect of the gas splitting, phase imprinting and free evolution performed in the experiments [10, 14, 15]. We implement these manipulations through the functions $I(t)$ and $F(t)$ which are present in the definition (3) of $V_\perp(y, t)$. We distinguish a number of stages:

1. A single gas is prepared in a thermal state. The transverse confining potential is a single well with a flat bottom, given by (3) with $I = I_c$ and $F = 0$.

2. We raise the double well barrier over some time $t_r$ by increasing $I$ linearly from $I_c$ to $I_{\max}$. At $t = t_r$ we are left with a split gas and a high tunnel barrier.

3. We raise one of the wells over a time $t_{\mathrm{imp}}$ by increasing $F(t)$ linearly from 0 to $F_{\max} > 0$. Physically, this imprints a phase difference between the wells.

4. We remove the imbalance between the wells by tuning $F(t)$ back down to zero in time $t_{\mathrm{imp}}$.

5. Finally we lower the tunnel barrier somewhat to enable tunneling on the relevant time scales, by decreasing $I$ from $I_{\max}$ to $I_f$ in a time $t_{\mathrm{low}}$.

In order to achieve these steps, we choose the functions $I(t)$ and $F(t)$ from Eq. (3) as

$$
I(t) = \begin{cases}
I_c + (I_{\max} - I_c)\frac{t}{t_r}\,, & \text{if } t < t_r\,, \\
I_{\max}\,, & \text{if } t_r \leq t < t_r + 2t_{\mathrm{imp}}\,, \\
I_{\max} + (I_f - I_{\max})\frac{t - t_r - 2t_{\mathrm{imp}}}{t_{\mathrm{low}}}\,, & \text{if } t_r + 2t_{\mathrm{imp}} \leq t < t_r + 2t_{\mathrm{imp}} + t_{\mathrm{low}}\,, \\
I_f\,, & \text{else}\,,
\end{cases} \tag{54}
$$

and

$$
F(t) = \begin{cases} 0 , & \text{if } t < t_\mathrm{r} , \\ \frac{t - t_\mathrm{r}}{t_\mathrm{imp}} , & \text{if } t_\mathrm{r} \le t < t_\mathrm{r} + t_\mathrm{imp} , \\ 1 - \frac{t - t_\mathrm{r} - t_\mathrm{imp}}{t_\mathrm{imp}} , & \text{if } t_\mathrm{r} + t_\mathrm{imp} \le t < t_\mathrm{r} + 2t_\mathrm{imp} , \\ 0 , & \text{if } t_\mathrm{r} + 2t_\mathrm{imp} \le t , \end{cases}
\tag{55}
$$

with $t_\mathrm{r} = 5\,\mathrm{ms}$, $t_\mathrm{imp} = t_\mathrm{low} = 2\,\mathrm{ms}$, $I_\mathrm{c} = 0.4$, $I_\mathrm{max} = 0.58$ and $I_\mathrm{f} = 0.5$.

## 5.2 Numerical determination of the initial state

At stage 1, the system is initialized in a thermal state of the Hamiltonian (12), subject to the HF approximation (50). This state is determined as follows. We expand the field operators as

$$
\hat{\psi}_a(x) = \sum_{a,\alpha} \chi_\alpha(x) \Phi_a(y) \hat{b}_{a\alpha} ,
\tag{56}
$$

where $\chi_\alpha$ are real eigenfunctions of the harmonic oscillator potential in the $x$-direction, and we keep $n_\mathrm{m} + 1$ such modes. The Hamiltonian (12) subject to (50) can then be written as

$$
H_{\mathrm{1D}}^{(\overline{a})}(0) = \sum_{a,b=0}^{\overline{a}-1} \sum_{\alpha,\beta=0}^{n_\mathrm{m}} h_{a\alpha,b\beta} \hat{b}_{a\alpha}^\dagger \hat{b}_{b\beta},
\tag{57}
$$

with the tensors

$$
h_{a\alpha,b\beta} = \delta_{a,b}\delta_{\alpha,\beta} \left[ \omega_x (\alpha + 1/2) + \epsilon_a(0) \right] + 4 \sum_{c,d=0}^{\overline{a}-1} \sum_{\gamma,\delta=0}^{n_\mathrm{m}} \Gamma_{abcd}(0) \overline{\Gamma}_{\alpha\beta\gamma\delta} \left\langle \hat{b}_{c\gamma}^\dagger \hat{b}_{d\delta} \right\rangle ,
$$

$$
\overline{\Gamma}_{\alpha\beta\gamma\delta} = \int dx\, \chi_\alpha(x)\chi_\beta(x)\chi_\gamma(x)\chi_\delta(x) .
\tag{58}
$$

Reshaping $h_{a\alpha,b\beta}$ and diagonalizing the resulting matrix numerically yields a canonical transformation

$$
\hat{b}_{a\alpha} = \sum_{b\beta} P_{a\alpha,b\beta} \hat{c}_{b\beta} .
\tag{59}
$$

The new creation and annihilation operators diagonalize the Hamiltonian $H_{\mathrm{1D}}^{(\overline{a})}(0) = \sum_{a\alpha} E_{a\alpha} \hat{c}_{a\alpha}^\dagger \hat{c}_{a\alpha}$. Assuming the $\hat{c}$'s to have thermal occupation numbers with respect to this Hamiltonian then gives

$$
\left\langle \hat{b}_{c\gamma}^\dagger \hat{b}_{d\delta} \right\rangle = \sum_{a\alpha} \frac{P_{c\gamma,a\alpha}^\dagger P_{a\alpha,d\delta}}{e^{(E_{a\alpha}-\mu)/k_\mathrm{B}T} - 1} ,
\tag{60}
$$

which combined with (58) forms a self-consistent system of equations. We proceed by iteration: starting from an initial guess $\left\langle \hat{b}_{c\gamma}^\dagger \hat{b}_{d\delta} \right\rangle_0$, which we take to be thermal with respect to the non-interacting Hamiltonian, we diagonalize $h_{a\alpha,b\beta}$ and compute (60) with the resulting $P$ and $E$. Reinserting into (58) leads to the next iteration, and we repeat until convergence is reached.

A major hurdle in the above procedure is presented by the overlap tensor $\overline{\Gamma}_{\alpha\beta\gamma\delta}$. As we use $n_\mathrm{m} = 1000$ modes, this tensor is too large to store numerically. However, using known identities for Hermite polynomials [77], we can write (58) as

$$
\overline{\Gamma}_{\alpha\beta\gamma\delta} = \sqrt{m\omega_x} \sum_{p=0}^{2n_\mathrm{m}} A_{\alpha\beta}^p A_{\gamma\delta}^p ,
\tag{61}
$$

$$
A_{\alpha\beta}^p = \sum_{m=0}^{\min(\alpha,\beta)} B_{\alpha\beta}^{pm} ,
\tag{62}
$$

where the tensors $B_{\alpha\beta}^{pm}$ are 0 if $\alpha + \beta - 2m - p$ is odd and/or negative, and otherwise given by

$$B_{\alpha\beta}^{pm} = \frac{m!}{\sqrt{\alpha!\beta!}} \frac{2^m}{\sqrt{2^{\alpha+\beta}}} \binom{\alpha}{m}\binom{\beta}{m} \frac{(\alpha+\beta-2m)!(-1/2)^{\frac{1}{2}(\alpha+\beta-2m-p)}}{\sqrt{p!}((\alpha+\beta-2m-p)/2)!} \ . \tag{63}$$

The considerably smaller tensors $A_{\alpha\beta}^{p}$ can now be separately contracted with other terms in (58), leading to a great memory gain. Even so, evaluating and storing the tensors (63) is still a very slow process for $n_{\mathrm{m}} = 1000$. We therefore make a simplifying assumption: we set

$$A_{\alpha\beta}^{p} \to 0 \quad \text{if} \quad |\alpha - \beta| > \Lambda \tag{64}$$

for some $\Lambda$, which we choose to be 40 in our numerics. To see how this is justified, we note that the Hamiltonian (57)-(58) implies the relation

$$[\omega_x(\alpha+1/2) + \epsilon_a(0) - E_{\overline{a}\overline{a}}]P_{a\alpha,\overline{a}\overline{\alpha}} =$$
$$= 4 \sum_{b,c,d} \sum_{\beta,\gamma,\delta} \Gamma_{abcd}(0)\overline{\Gamma}_{\alpha\beta\gamma\delta} \sum_{\overline{c}\overline{\gamma}} \frac{P_{c\gamma,\overline{c}\overline{\gamma}}^{\dagger}P_{\overline{c}\overline{\gamma},d\delta}}{e^{(E_{aa}-\mu)/k_{\mathrm{B}}T} - 1} P_{b\beta,\overline{a}\overline{\alpha}} \tag{65}$$

on the canonical transformations $P$ for all $a, \overline{a}, \alpha, \overline{\alpha}$. The assumption (64) is therefore valid if the $P_{a\alpha,b\beta}$ become very small whenever $|\alpha - \beta| \gtrsim \Lambda$. This is reasonable since the weak interactions are not expected to couple harmonic oscillator modes that have widely different numbers of nodes. We check *a posteriory* that this assumption is consistent and well within the range set by $\Lambda$. We have also checked the assumption explicitly for the case of $n_{\mathrm{m}} = 400$. Finally, we have verified that the Green's functions resulting from the above procedure remain time-independent when they are propagated in time under (51) with a *time-independent* potential $V_\perp(y, 0)$.

The above procedure yields a set of Green's functions $C_{ij}(x, y)$ which characterize the state of the system at $t = 0$. In the central region of the trap, with $|x| < 3\,\mu$m, we find exponential decay of the Green's functions $C_{ii}(x, -x)$ for the parameters presented in Sec. 6.1. The associated correlation length is roughly $0.5\,\mu$m.

# 6 Josephson oscillations

We are now in a position to model the full experimental sequence. To do so, we first fix the values for various constants and parameters.

## 6.1 Experimental parameters

The transverse potential $V_\perp(y, t)$ is described by Eq. (3) and its time evolution follows Sec. 5.1 with $t_{\mathrm{r}} = 4\,$ms, $t_{\mathrm{imp}} = 2\,$ms and $t_{\mathrm{low}} = 2\,$ms. This means that after a time $t_{\mathrm{r}} + 2t_{\mathrm{imp}} + t_{\mathrm{low}} = 11\,$ms, the confining potential becomes time-independent, and the 1D field operators lose their explicit time-dependence as a result. We consider a temperature of 60 nK and take the transverse confining potential in the $z$-direction to be harmonic with $\omega_z = 2\pi \cdot 1.7\,$kHz. The s-wave scattering length and atomic mass for the experimental system of ${}^{87}$Rb atoms [15] are $a_{\mathrm{s}} \approx 5.2\,$nm and $m \approx 1.4 \cdot 10^{-25}$kg, respectively. This fixes all parameters in the problem.

## 6.2 Assessment of time-dependent truncation errors in a toy model

In our full model, the initial thermal state contains three different transverse levels which mutually interact. An example of the resulting initial density profiles is given in Fig. 5(a), with

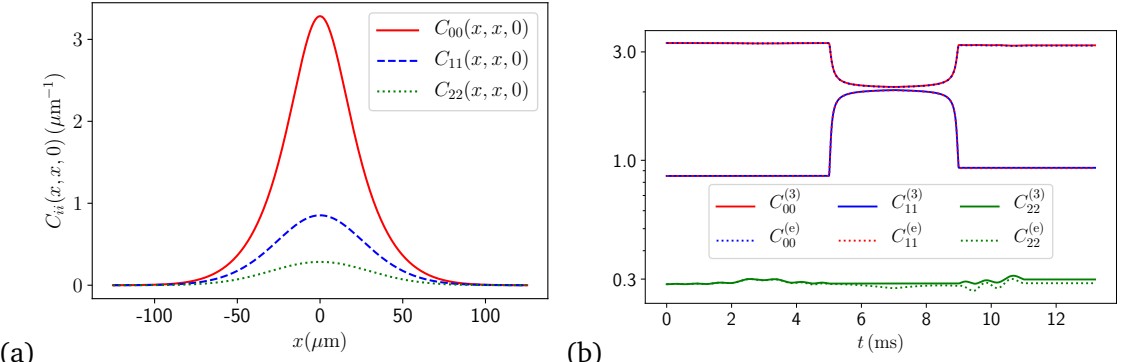

Figure 5: (a) Initial density profiles of levels $0, 1, 2$ at $T = 60\,\text{nK}$, $\omega_x = 2\pi \cdot 12.5\,\text{Hz}$ and $N = 259$. (b) Time evolution of Green's functions $C_{ii} = \langle \hat{\psi}_i^\dagger \hat{\psi}_i \rangle$ for the quantum mechanical problem of noninteracting bosons in a double well. This corresponds to the PDE (51) in the absence of $x$-dependence and with $\Gamma_{ijkl} = 0$. We compare the problem with truncation index $\bar{a} = 3$ (as in the full model, solid curves) to results for $\bar{a} = 15$ (dotted curves). The latter is chosen by looking for convergence in $\bar{a}$. The initial conditions match the peak densities from panel (a) at $t = 0$ and a vertical log-scale is chosen to highlight changes in $C_{22}$.

occupation of the higher levels being suppressed as expected thanks to their larger energy cost. For $t > 0$, the occupations can change in a way that is both due to interactions and to the non-adiabaticity of the deformation of $V_\perp(y, t)$. The latter is modelled by the additional term (49) in the equations of motion (45), which are truncated at $a = \bar{a} = 3$. To assess the error made in this truncation, we briefly consider the quantum mechanical problem of bosons in a double well $V_\perp(y, t)$. We discard the $x$-direction and set interactions to zero, so that the problem is given by Eq. (51) in the absence of $x$-dependence and with $\Gamma_{ijkl} = 0$. This problem can be integrated numerically for any value of the truncation index $\bar{a}$. Results for $\bar{a} = 3$ (as we use in the full model) and $\bar{a} = 15$ are compared in Fig. 5(b). The lines remain close, showing that the truncation error has a very small effect on transitions induced by the time-dependence of $V_\perp(y, t)$.

## 6.3 Characterization of the quantum state after the preparation sequence

In our Hartree–Fock approximation, the state of the system at time $t$ is fully determined by the Green's functions $C_{ij}(x, y, t)$, with $i, j = 0, \dots, \bar{a} - 1$. Having these at hand thus allows us to give a full, quantitative description of the state of the system after the splitting and phase imprinting procedure, at the level of Hartree–Fock. This is a major improvement beyond existing, more phenomenological [67, 68] or approximate [69, 70] methods. As an illustration of the ability of our method to provide the full Green's functions, we plot $C_{00}(x, y, t)$ and $|C_{01}(x, y, t)|$ at time $t = t_r + 2t_{\text{imp}} + t_{\text{low}} = 11\,\text{ms}$, that is, after the preparation sequence (see Fig. 6). One sees that the Green's functions are strongly peaked around the main diagonal. To further illustrate their behavior, it is therefore instructive to plot the diagonal (Fig. 7) and anti-diagonal (Fig. 8) of the Green's functions of interest.

An equivalent way to express the same Green's functions is through the occupation numbers

$$M_{\alpha,\beta}^{(ab)} \equiv \left\langle \hat{b}_{a\alpha}^\dagger \hat{b}_{b\beta} \right\rangle, \tag{66}$$

where $\hat{b}_{a\alpha}^\dagger$ creates a particle in instantaneous eigenstate $\xi_\alpha(x)\Phi_a(y, t)$, as defined via Eq. (56). As the occupation numbers are strongly suppressed away from the diagonal, we display this

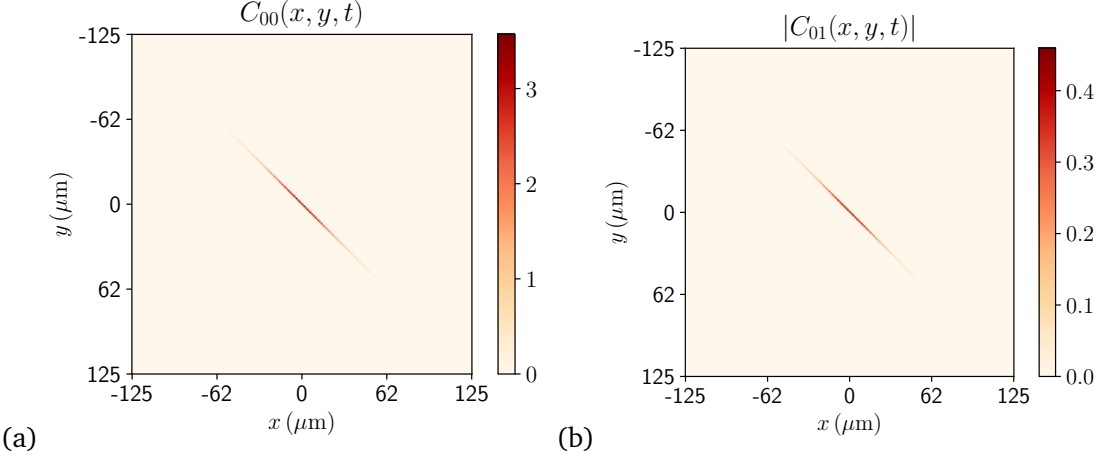

(a)

(b)

Figure 6: Sample pictures for the Green's functions at time $t = t_\mathrm{r} + 2t_\mathrm{imp} + t_\mathrm{low} = 11\,\mathrm{ms}$, that is, after the preparation sequence. The parameters are as described in Section 6.1, with $\omega_x = 2\pi \cdot 12.5\,\mathrm{Hz}$, $T = 60\,\mathrm{nK}$ and $N = 259$ particles.

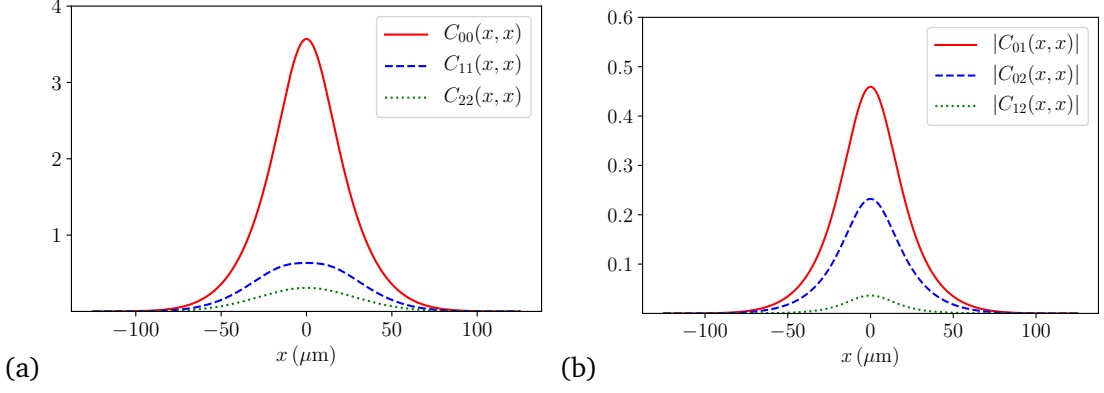

(a)

(b)

Figure 7: The spatial diagonal of Green's functions $C_{ij}(x, y)$ of interest, with the same parameters as in Fig. 6.

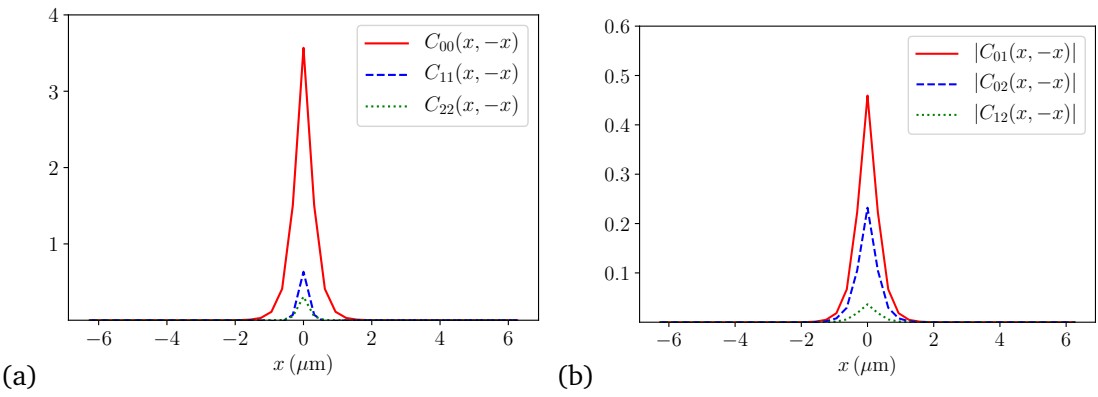

(a)

(b)

Figure 8: The spatial anti-diagonal of Green's functions $C_{ij}(x, y)$ of interest, with the same parameters as in Fig. 6.

diagonal (Fig. 10) and the anti-diagonal pertaining to $\alpha + \beta = 20$ (Fig. 9) for all occupation numbers of interest.

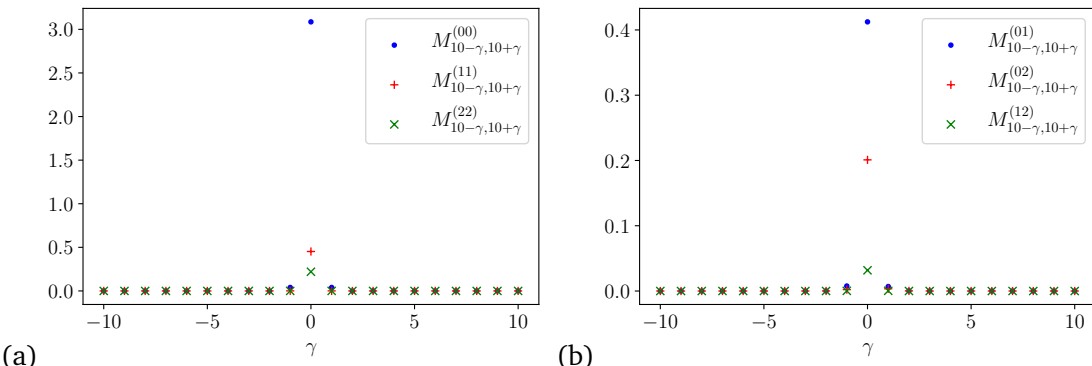

Figure 9: Anti-diagonal cut of occupation numbers $M_{\alpha,\beta}^{(ab)}$, defined in Eq. (66). The cut corresponds to $\alpha = 10 + \gamma$ and $\beta = 10 - \gamma$. The chosen parameters are the same as in Fig. 6.

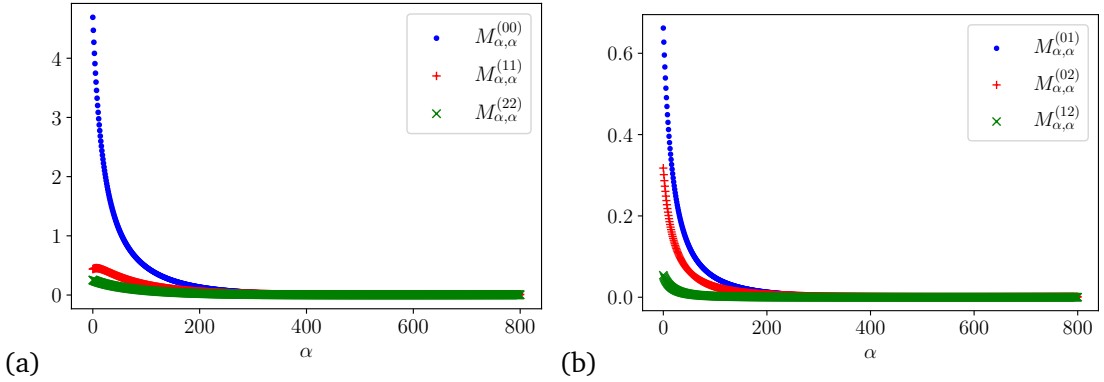

Figure 10: The diagonal elements ($\alpha = \beta$) of occupation numbers $M_{\alpha,\beta}^{(ab)}$, defined in Eq. (66). The chosen parameters are the same as in Fig. 6.

In a nutshell, the preparation sequence described above provides us with an initial state characterized by very short-ranged correlations. In the centre of the trap the correlation length is roughly $0.5\mu$m, in line with the correlation length at $t = 0$.

## 6.4 Damping of density-phase oscillations

By monitoring the observables from Sec. 3, we can follow the relative density and phase between the gases. As soon as the barrier is lowered (step 5. in Sec. 5), oscillations in the relative density and phase can be observed, *cf.* Fig. 11(a), with an offset of a quarter period between the two. Importantly, the amplitude shows an initial period of damping, for all particle numbers we have considered. The mean interference contrast $\mathcal{C}(x,t)$, on the other hand, shows only very limited time-dependence. We have fitted the density-phase oscillations at the center of the trap between $t = 11$ ms and $t = 35$ ms to

$$\varphi(t) = e^{-t/\tau} \sin(\omega t + \varphi_0), \tag{67}$$

and extracted the damping time $\tau$ and frequency $\omega$. We stress that this is by no means a full description of the phase oscillations but merely a phenomenological formula to quantify the

time scale $\tau$ of the damping observed in the early oscillation stage of the HF simulation. The dependence of this damping time $\tau$ on $N$ is displayed in Fig. 11(b), whereas the dependence of the frequency $\omega$ on $N$ is displayed in Fig. 12. There is a range of values of $N$ for which the

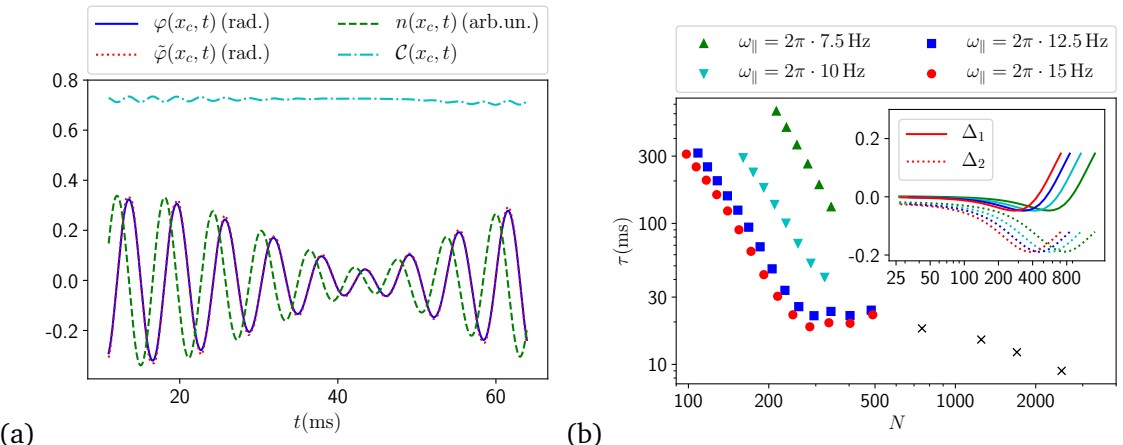

(a)  (b)

Figure 11: (a) oscillations of relative density $n$ and phase $\varphi$ in the center of the trap ($x = x_c$) for $T = 60$ K, $N = 259$ and $\omega_x = 2\pi \cdot 12.5$ Hz. $\tilde{\varphi}$ denotes the relative phase computed using Choice 2 from Sec. 3. The mean interference contrast $\mathcal{C}(x, t)$ from Eq. (39) is also plotted, and is almost constant in time. (b) Big colored dots: damping times extracted from a fit with Eq. (67). Black dots: damping times reported in [10]. Inset: reproduction of Fig. (4), showing errors $\Delta_{1,2}$ between HF and YY+LDA from Eq. (53) for $T = 60$ K.

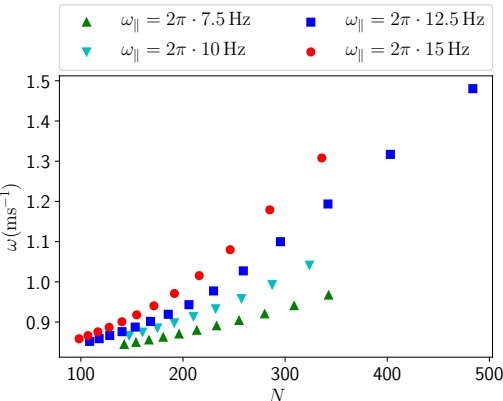

Figure 12: Frequencies $\omega$ of density-phase oscillations at the center of the trap as a function of the number of particles in the gas, $N$. The frequencies are extracted from a fit with Eq. (67). The temperature of the initial state is 60 nK.

damping time as a function of $N$ is in qualitative agreement with the power-law dependence reported in [10,14]. For $N \sim 300$, the behavior suddenly changes. This transition coincides with the breakdown of HF in the initial state: around this particle number, the errors $\Delta_{1,2}$ between HF and YY+LDA from Eq. (53) start to increase to significant values. This is displayed in the inset to Fig. 11(b). We thus conjecture that the deviation of $\tau(N)$ from a power law for $N \gtrsim 300$ is due to a breakdown of HF in that regime.

A number of additional observations can be made.

- After showing a damped oscillatory behaviour up to times of $t \approx 40$ms the Josephson

oscillations begin to increase again. This effect is not observed in the experiments, which as we have stressed throughout have been performed in a different parameter regime not accessible by HF. We note however, that the experiments focused on time scales of below $40-60$ms, so that it cannot be ruled out that at later times a reemergence of the oscillations occurs in the experimentally relevant parameter regime as well. It would be interesting to repeat the experiments for lower particle numbers in order to study this reemergence in detail.

- The frequency of density-phase oscillations is highest at the center $x_c$ of the trap in the $x$-direction. Away from this point, the frequency is smaller, as displayed in Fig. 13(a). This figure also shows that the damping during the first few periods is somewhat weaker

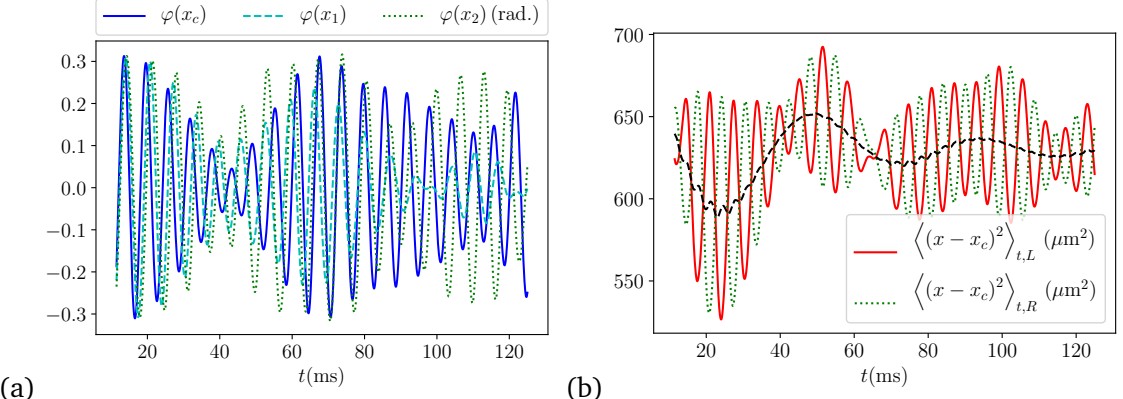

(a)

(b)

Figure 13: Additional plots for the same parameters as Fig. 11(a). (a) relative phase at the trap center ($x = x_c$) and at positions $x_1 = x_c + 25\,\mu$m, $x_2 = x_c + 37.5\,\mu$m. (b) squared longitudinal size (68) of left and right gases (*red and green*) as well as their average (*black*).

at points away from the trap center, where the gas density is smaller.

- The gas as a whole shows a breathing motion. This can be shown by studying the squared longitudinal size of the left and right gas profiles,

$$\left\langle (x - x_c)^2 \right\rangle_{t,i} \equiv \int dx\, C_{ii}(x,x,t)(x-x_c)^2 \Big/ \int dx\, C_{ii}(x,x,t), \quad i = L,R. \tag{68}$$

Fig. 13(b) shows that the squared longitudinal sizes of the left and right gases oscillate out of phase with one another. On top of this, there is an overall breathing motion of the gas with a frequency that depends monotonously on $\omega_x$. This breathing gets damped over a timescale that is large compared to the breathing period of the separate left and right gases.

- The time scale of the breathing motion of the gas is seen to coincide with the time at which the Josephson oscillations reemerge after the initial oscillatory decay.

It is instructive to investigate the effect on the damping that various aspects of our set-up might have. First, there are two possible definitions of left- and right-localized bosons $\hat{\psi}_{L,R}$, as described in Sec. 3. As mentioned there, we stick to Choice 1 (*cf.* (36)) by default. Do our results, and the observed damping in particular, change if we switch to Choice 2? Fig. 11(a) shows results for Choice 2 in red. The curve is shown to lie very close to the blue curve, which was computed with Choice 1. This behavior occurred for all performed simulations, showing that the choice between Choices 1 and 2 does not significantly affect our results.

Second, we can investigate the effect of the second excited level, by turning off the corresponding couplings (13), setting $\Gamma_{2jkl} = 0$ for all permutations of indices. This completely shields the lowest two levels 0 and 1, and hence the relative density and phase (37), from any effects which level 2 might have. The resulting curves for $\varphi$ fall on top of the curves for nonzero interaction with the second excited level, as exemplified by Fig. 14(a). We conclude that the effect of the additional boson species on the damping is negligible.

Third, we can study the effect of the longitudinal potential on the damping. This effect turns out to be very significant. In Fig.11(b), we see that the $\tau(N)$-curves are shifted upwards as the strength of the potential is decreased. A weaker potential thus leads to a decrease in the damping effect. This suggests that in a box potential, the damping effect might be completely absent (within the SCHF approximation). We have therefore performed the same simulations in a box potential, by imposing hard wall boundary conditions at $x = x_c \pm L/2$ on the PDE (51). Fig. 14(b) shows a representative result, with parameters that are chosen to closely match those of Fig. 11(a). In particular, the bulk density is chosen to match the peak density from the initial condition of Fig. 11(a). The result is striking: in the box, no damping is visible at all. In fact, a very slight *increase* in the amplitude of the density-phase oscillations is observed.

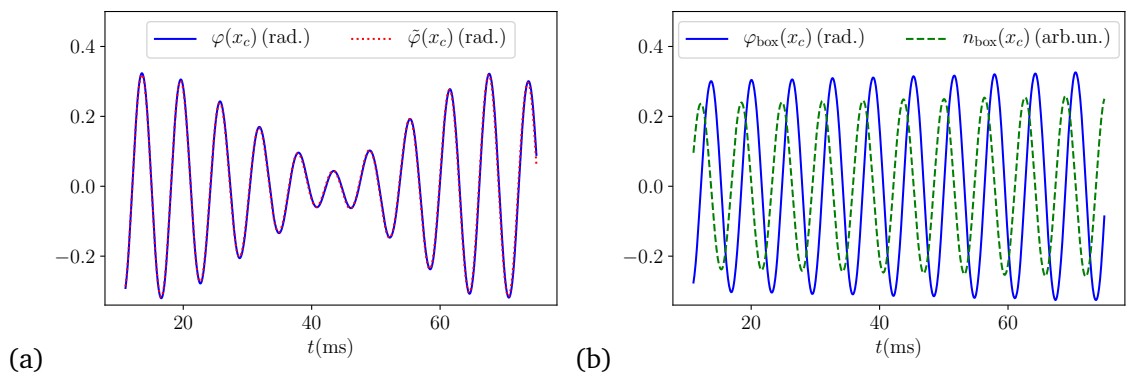

(a)          (b)

Figure 14: (a) the same curve as the phase $\varphi$ from Fig. 11(a), presented alongside the same quantity, but computed with $\Gamma_{2jkl} = 0$ for all permutations of indices. (b) oscillations of relative density $n$ and phase $\varphi$ for the same parameters as Fig. 11(a) but in a hard-wall box potential of size $L = 80\,\mu$m. The bulk density is chosen to match the peak density from the initial condition of Fig. 11(a)

# 7 Beyond self-consistent Hartree–Fock

The main attraction of the self-consistent Hartree–Fock approximation is its simplicity. However, it is not expected to provide a quantitatively accurate account of the non-equilibrium dynamics and it would be interesting to improve on it. A good way forward would be to implement the second Born approximation [82] following Refs. [83, 84]. The two significant complications compared to these works are the absence of translational invariance and the explicit time-dependence of the Hamiltonian during the splitting and phase imprinting sequence. As a first step we consider the non-equilibrium evolution after the phase imprinting, which is

described by a time-independent Hamiltonian (12)

$$\mathcal{H}_{1D} = \sum_{a=0}^{\bar{a}-1} \int dx \, \hat{\psi}_a^\dagger(x) \left[ -\frac{1}{2m}\frac{\partial^2}{\partial x^2} + \frac{m\omega^2}{2}x^2 + \epsilon_a \right] \hat{\psi}_a(x)$$

$$+ \int dx \sum_{a,b,c,d=0}^{\bar{a}-1} \Gamma_{abcd} \, \hat{\psi}_a^\dagger(x)\hat{\psi}_b^\dagger(x)\hat{\psi}_c(x)\hat{\psi}_d(x) \,. \tag{69}$$

We now expand in harmonic oscillator modes notation

$$\psi_a(x) = \sum_j \chi_j(x) b_{a,j} \,, \tag{70}$$

and substitute this back into the expression for the Hamiltonian. Introducing a multi-index

$$\boldsymbol{k} \equiv (a,j) \,, \quad b_{a,j} = b(\boldsymbol{k}) \,, \tag{71}$$

we can rewrite the Hamiltonian in a very compact form

$$\mathcal{H}_{1D} = \sum_{\boldsymbol{k}} \varepsilon(\boldsymbol{k}) b^\dagger(\boldsymbol{k}) b(\boldsymbol{k}) + \sum_{\boldsymbol{k}_1,\boldsymbol{k}_1,\boldsymbol{k}_3,\boldsymbol{k}_4} V(\boldsymbol{k}_1,\boldsymbol{k}_2,\boldsymbol{k}_3,\boldsymbol{k}_4) \, b^\dagger(\boldsymbol{k}_1)b^\dagger(\boldsymbol{k}_2)b(\boldsymbol{k}_3)b(\boldsymbol{k}_4). \tag{72}$$

Here we have defined

$$\varepsilon(\boldsymbol{k}) = \epsilon_a + \epsilon_j \,, \quad V(\boldsymbol{k}_1,\boldsymbol{k}_2,\boldsymbol{k}_3,\boldsymbol{k}_4) = \Gamma_{abcd}\bar{\Gamma}_{ijkl} \,, \tag{73}$$

where $\Gamma_{abcd}$ and $\bar{\Gamma}_{ijkl}$ are given by (13) and (58) respectively and $\boldsymbol{k}_1 = (a,i)$, $\boldsymbol{k}_2 = (b,j)$, $\boldsymbol{k}_3 = (c,k)$ and $\boldsymbol{k}_4 = (d,l)$. The second Born approximation for the single-particle Green's function

$$G(\boldsymbol{k},\boldsymbol{p},t) = \langle \Psi(t)|c^\dagger(\boldsymbol{k})\, c(\boldsymbol{p})|\Psi(t)\rangle \,, \tag{74}$$

can then be derived by generalizing the steps given in [84,85] to the case at hand. This results in the following set of equations of motion

$$\frac{\partial G(\boldsymbol{k},\boldsymbol{p},t)}{\partial t} = i\big(\varepsilon(\boldsymbol{k})-\varepsilon(\boldsymbol{p})\big)G(\boldsymbol{k},\boldsymbol{p},t)$$

$$+ 2i \sum_{\boldsymbol{q}_1,\ldots,\boldsymbol{q}_4} Y(\boldsymbol{k},\boldsymbol{p};\boldsymbol{q}_1,\ldots,\boldsymbol{q}_4)e^{itE(\boldsymbol{q}_1,\ldots,\boldsymbol{q}_4)}G(\boldsymbol{q}_1,\boldsymbol{q}_3,0)G(\boldsymbol{q}_2,\boldsymbol{q}_4,0)$$

$$- \int_0^t ds \sum_{\boldsymbol{q}_1,\ldots,\boldsymbol{q}_4} K(\boldsymbol{k},\boldsymbol{p};\boldsymbol{q}_1,\ldots,\boldsymbol{q}_4|t-s)\, G(\boldsymbol{q}_1,\boldsymbol{q}_2,s)G(\boldsymbol{q}_3,\boldsymbol{q}_4,s)$$

$$- \int_0^t ds \sum_{\boldsymbol{q}_1,\ldots,\boldsymbol{q}_6} L(\boldsymbol{k},\boldsymbol{p};\boldsymbol{q}_1,\ldots,\boldsymbol{q}_6|t-s)\, G(\boldsymbol{q}_1,\boldsymbol{q}_2,s)G(\boldsymbol{q}_3,\boldsymbol{q}_4,s)G(\boldsymbol{q}_5,\boldsymbol{q}_6,s) \,, \tag{75}$$

where $E(\boldsymbol{k}_1,\ldots,\boldsymbol{k}_4) = \varepsilon(\boldsymbol{k}_1) + \varepsilon(\boldsymbol{k}_2) - \varepsilon(\boldsymbol{k}_3) - \varepsilon(\boldsymbol{k}_4)$ and the integral kernels are given by

$$Y(\boldsymbol{k},\boldsymbol{p};\boldsymbol{k}_1,\boldsymbol{k}_2,\boldsymbol{k}_3,\boldsymbol{k}_4) = \Gamma(\boldsymbol{k}_1,\boldsymbol{k}_2,\boldsymbol{k}_3,\boldsymbol{k})\delta_{\boldsymbol{k}_4,\boldsymbol{p}} + \Gamma(\boldsymbol{k}_1,\boldsymbol{k}_2,\boldsymbol{k},\boldsymbol{k}_4)\delta_{\boldsymbol{k}_3,\boldsymbol{p}}$$

$$- \Gamma(\boldsymbol{p},\boldsymbol{k}_2,\boldsymbol{k}_3,\boldsymbol{k}_4)\delta_{\boldsymbol{k}_1,\boldsymbol{k}} - \Gamma(\boldsymbol{k}_1,\boldsymbol{p},\boldsymbol{k}_3,\boldsymbol{k}_4)\delta_{\boldsymbol{k}_2,\boldsymbol{k}}, \tag{76}$$

$$L(\boldsymbol{k},\boldsymbol{p};\boldsymbol{q}_1,\ldots,\boldsymbol{q}_6|t) = 8\sum_{\boldsymbol{p}} X(\boldsymbol{k},\boldsymbol{p};\boldsymbol{q}_1,\boldsymbol{q}_3,\boldsymbol{q}_6,\boldsymbol{p};\boldsymbol{p},\boldsymbol{q}_5,\boldsymbol{q}_2,\boldsymbol{q}_4|t)$$

$$+ 16\sum_{\boldsymbol{p}} X(\boldsymbol{k},\boldsymbol{p};\boldsymbol{q}_1,\boldsymbol{q}_3,\boldsymbol{q}_2,\boldsymbol{p};\boldsymbol{p},\boldsymbol{q}_5,\boldsymbol{q}_4,\boldsymbol{q}_6|t) \,,$$

$$K(\boldsymbol{k},\boldsymbol{p};\boldsymbol{q}_1,\boldsymbol{q}_2,\boldsymbol{q}_3,\boldsymbol{q}_4|t) = 8\sum_{\boldsymbol{k}_1,\boldsymbol{k}_2} X(\boldsymbol{k},\boldsymbol{p};\boldsymbol{k}_1,\boldsymbol{k}_2,\boldsymbol{q}_2,\boldsymbol{q}_4;\boldsymbol{q}_1,\boldsymbol{q}_3,\boldsymbol{k}_1,\boldsymbol{k}_2|t) \,,$$

$$X(\boldsymbol{k},\boldsymbol{p};\boldsymbol{q}_1,\boldsymbol{q}_2,\boldsymbol{q}_3,\boldsymbol{q}_4;\boldsymbol{k}_1,\boldsymbol{k}_2,\boldsymbol{k}_3,\boldsymbol{k}_4) = \Gamma(\boldsymbol{q}_1,\boldsymbol{q}_2,\boldsymbol{q}_3,\boldsymbol{q}_4)e^{iE(\boldsymbol{q}_1,\boldsymbol{q}_2,\boldsymbol{q}_3,\boldsymbol{q}_4)}Y(\boldsymbol{k},\boldsymbol{p};\boldsymbol{k}_1,\boldsymbol{k}_2,\boldsymbol{k}_3,\boldsymbol{k}_4)$$

$$- \{\boldsymbol{q}_j \leftrightarrow \boldsymbol{k}_j\}. \tag{77}$$

The set of integro-differential equations (75) is clearly much more difficult to solve numerically than the self-consistent Hartree–Fock equations. The time integration is crucial at short times, while for sufficiently late times (75) ought to be reducible to a matrix quantum Boltzmann equation [84]. Integrating (75) is beyond the scope of this paper, but some general comments are in order. It is clear that in order to be able to integrate (75) numerically only a limited number of different **k** modes can be retained. Hence one should focus on the case where the longitudinal confinement is fairly tight. In this (experimentally readily accessible) case interaction effects beyond the SCHF approximation can be analyzed through (75).

## 8 Conclusions

In this work we have developed a microscopic theory for the non-equilibrium evolution of bosons confined by a time-dependent quasi-one-dimensional trapping potential. Using that the transverse confinement is tight we have projected the full three-dimensional theory to a finite number of coupled, one-dimensional channels. By employing a time-dependent projection the number of channels that need to be retained in experimentally relevant parameter regimes is very small: three channels suffice. We then analyzed the resulting theory by means of a self-consistent time-dependent Hartree–Fock approximation and showed how the resulting Green's functions are related to averages of experimentally measured quantities. The Hartree–Fock approximation is expected to apply only for sufficiently weak interactions and sufficiently high energy densities. We have tried to identify a corresponding parameter regime by comparing the SCHF approximation to results obtained by combining the exact solution of the Lieb-Liniger model with a local density approximation in the trapping potential. On the basis of these considerations we restricted our initial states to temperatures of at least 60 nK and to particle numbers below $\sim 200$. In this parameter regime we expect the HF method to work well at least at short times, when the neglected higher connected $n$-point functions have not had time to grow substantially.

Our method has a number of attractive features. First, it allows to include the effects of various longitudinal potentials. Second, it can account for higher excited levels of the transverse confining potential which are normally neglected. Finally, it allows us to model the gas splitting and phase imprinting in a fully microscopic way. To our knowledge, such a model has not been presented before, and one of our main results is a characterization of the quantum state of the system after gas splitting and phase imprinting in terms of single-particle Green's functions of the one-dimensional channels. The second main result of our work is the description of the density-phase oscillations that ensue after the splitting and phase imprinting. In particular we find that these are damped over a few oscillation periods. These damped oscillations agree with recent measurements [10, 14, 15] in multiple ways. First, the damping time is inversely related to the number of particles, following a curve compatible with [10]. Second, the oscillation frequency decreases away from the center of the trap, as observed in [15]. We have shown that the coupling to the second excited level has very little effect on the damping. On the other hand, the longitudinal trapping potential is seen to play a very important role: the weaker the longitudinal trapping frequency, the weaker the damping. In a hard wall box, no damping is observed at all within our Hartree–Fock approximation. This suggests that damping effects are suppressed in this geometry for weak interactions. It therefore would be very interesting to repeat the experiments [10, 14, 15] in a hard-wall box potential. Such potentials are indeed under development [12, 16] and our model can serve as a direct theoretical prediction for such setups.

The main limitation of our method is the way interactions are treated. In order to access the parameter regime of the experiments [10, 14, 15], in which the particle number was

significantly higher than in our simulations, it is necessary to go beyond the SCHF approximation used here. A major improvement can be provided by the second Born approximation discussed in section 7, but this is much harder to implement numerically. Ideally one would want to employ a controlled approximation scheme like [86,87] for our fully time-dependent problem.

Our work has several implications for attempts to describe Josephson oscillations in tunnel-coupled one-dimensional Bose gases based on the sine-Gordon model. Firstly, our work suggests that the experimental protocol for splitting and phase imprinting does not lead to a strong population of higher transverse levels as long as the effective temperature of the initial thermal state is sufficiently low. This implies that a description in terms of a low-energy effective field theory based on a sine-Gordon model with appropriate perturbations should apply. There are several kinds of perturbations that should be considered. A key finding of our work is the strong effect the longitudinal confining potential has on the damping of Josephson oscillations in the parameter regime studied here. This suggests that the low-energy field theory calculations based on the sine-Gordon model [62, 64–66] should be extended to account for the longitudinal confinement. This is certainly possible in the framework of the self-consistent time-dependent harmonic approximation used in [62,66]. Apart from the confining potential there are other perturbations to the sine-Gordon model that should be analysed. In particular one should consider the effects of the nonlinearities that arise from the curvature terms in the kinetic energy of the split Bose gas. These are formally irrelevant in equilibrium but could well play an important role in non-equilibrium dynamics.

Secondly, our characterization of the "initial state" after splitting and phase imprinting provides very useful information on what initial states to consider in the sine-Gordon framework. In the first instance one should consider Gaussian states with very short correlation lengths that reproduce the single-particle Green's functions reported here. Our microscopic modelling of the splitting process enables us to provide the same kind of information also in previously studied cases without phase-imprinting.

## Acknowledgements

We are grateful to the Erwin Schrödinger International Institute for Mathematics and Physics for hospitality and support during the programme on *Quantum Paths*. This work was supported by the EPSRC under grant EP/S020527/1 and YDvN is supported by the Merton College Buckee Scholarship and the VSB, Muller and Prins Bernhard Foundations. JS acknowledges the support by the Austrian Science Fund (FWF) via the DFG-FWF SFB 1225 ISOQUANT (I 3010-N27).

## A  Low energy projection in equilibrium

For simplicity we consider a two-dimensional system with time-independent Hamiltonian

$$H = \int dx \, dy \left\{ \Psi^{\dagger}(x,y) \left[ -\frac{\nabla^2}{2m} + \frac{m\omega^2}{2}x^2 + V_{\perp}(y) \right] \Psi(x,y) + c \left( \Psi^{\dagger}(x,y) \right)^2 \left( \Psi(x,y) \right)^2 \right\}. \quad (78)$$

The quadratic part can be diagonalized by going to a basis of single-particle eigenstates

$$\Psi(x,y) = \sum_{j,k=0}^{\infty} \chi_j(x) \Phi_k(y) \, b_{j,k} \, . \quad (79)$$

Here $\chi_j(x)$ are harmonic oscillator wave functions and $\Phi_k(y)$ are orthonormal eigenstates of the Hamiltonian $H_y$

$$H_y = -\frac{1}{2m}\frac{d^2}{dy^2} + V_\perp(y), \quad H_y\Phi_k(y) = \varepsilon_k\Phi_k(y). \tag{80}$$

In terms of the new canonical Bose fields

$$\Psi_k(x) = \int dy\, \Phi_k^*(y)\Psi(x,y), \quad [\Psi_j(x),\Psi_k(x')] = \delta_{j,k}\delta(x-x') \tag{81}$$

the Hamiltonian becomes

$$H = \int dx \sum_{k=0}^{\infty} \Psi_k^\dagger(x)\, h_k\, \Psi_k(x) + \int dx \sum_{k_1,k_2,k_3,k_4=0}^{\infty} V_{k_1,k_2,k_3,k_4}\Psi_{k_1}^\dagger(x)\Psi_{k_2}^\dagger(x)\Psi_{k_3}(x)\Psi_{k_4}(x). \tag{82}$$

Here we have defined

$$h_k = \left[-\frac{1}{2m}\frac{\partial^2}{\partial x^2} + \frac{m\omega^2}{2}x^2 + \varepsilon_k\right],$$

$$V_{k_1,k_2,k_3,k_4} = c\int dy\,\Phi_{k_1}^*(y)\Phi_{k_2}^*(y)\Phi_{k_3}(y)\Phi_{k_4}(y). \tag{83}$$

The imaginary time path integral representation of the partition function is

$$Z(\beta) = \int \prod_{k=0}^{\infty} \mathcal{D}\psi_k^*(\tau,x)\,\mathcal{D}\psi_k(\tau,x)\,e^{-S[\psi_n^*,\psi_n]}, \tag{84}$$

where

$$S[\psi_n^*,\psi_n] = \int_0^\beta d\tau \int dx \left\{ \sum_{k=0}^{\infty} \psi_k^*(\tau,x)\left[\frac{\partial}{\partial\tau} + h_k\right]\psi_k(\tau,x) \right. \tag{85}$$

$$\left. + \sum_{k_1,k_2,k_3,k_4=0}^{\infty} V_{k_1,k_2,k_3,k_4}\,\psi_{k_1}^\dagger(\tau,x)\psi_{k_2}^\dagger(\tau,x)\psi_{k_3}(\tau,x)\psi_{k_4}(\tau,x) \right\}.$$

The situation we are interested in is where the eigenvalues $\epsilon_k$ of the transverse confining potential constitute a large energy scale and the transverse level spacings $|\varepsilon_k - \varepsilon_j|$ between highly excited transverse states are large too. We can then "integrate out" the transverse degrees of freedom above some cutoff $\Lambda$. Let us denote the first eigenvalue above $\Lambda$ by $\epsilon_{\bar{a}}$, and rewrite the action as

$$S[\psi_n^*,\psi_n] = S_< + S_> + S_{\text{int}}, \tag{86}$$

where $S_<$ is the part of the action that only involve the fields $\Psi_k, \Psi_k^\dagger$ with $0 \le k < \bar{a}$, $S_>$ the quadratic part of the action that involves only fields with $k \ge \bar{a}$, and $S_{\text{int}}$ are the remaining quartic terms that mix channels below and above the cutoff and describe interactions between channels above the cutoff. Defining

$$\langle \mathcal{O}\rangle_> = \int \prod_{k=\bar{a}}^{\infty} \mathcal{D}\psi_k^*(\tau,x)\,\mathcal{D}\psi_k(\tau,x)\,\mathcal{O}\,e^{-S_>}, \tag{87}$$

we can eliminate the degrees of freedom above the cutoff using that we are dealing with weak interactions. Up to second order in $S_{\text{int}}$ we have the following expression for the low-energy part of the action

$$S_{\text{eff}} = S_< + \langle S_{\text{int}}\rangle_> - \frac{1}{2}\left[\langle S_{\text{int}}^2\rangle_> - \langle S_{\text{int}}\rangle_>^2\right]. \tag{88}$$

The first order term generates hopping between the low-energy channels

$$\langle S_{\text{int}} \rangle_> = \int_0^\beta d\tau \int dx \sum_{k_1,k_2=0}^{\bar{a}-1} W_{k_1,k_2}\, \psi_{k_1}^\dagger(\tau,x)\psi_{k_2}(\tau,x)\,, \tag{89}$$

where

$$W_{k_1,k_2} = \sum_{n=\bar{a}}^\infty 4V_{n,k_1,n,k_2} \sum_{j=0}^\infty \frac{|\chi_j(x)|^2}{e^{\beta\left(\varepsilon_k+\omega(j+1/2)\right)}-1}\,. \tag{90}$$

We see that this is small compared to the interaction strength $c$ because the Bose occupation factors are by construction negligible. The second order term in $S_{\text{int}}$ contains all possible quadratic, quartic and sextic interactions involving $\psi_k(x,\tau)$ and $\psi_k^*(x,\tau)$ compatible with particle number conservation, e.g.

$$\sum_{k_1,k_2,k_3,k_4=0}^{\bar{a}-1} \int d\tau \int d\tau' \int dx \int dx'\, U_{k_1,k_2,k_3,k_4}(\tau-\tau',x,x')$$
$$\times\ \psi_{k_1}^*(\tau,x)\psi_{k_2}(\tau,x)\psi_{k_3}^*(\tau',x')\psi_{k_4}(\tau',x')\,, \tag{91}$$

where

$$U_{k_1,k_2,k_3,k_4}(\tau-\tau',x,x') = -8 \sum_{n_2,n_2=\bar{a}}^\infty V_{n_1,k_1,n_2,k_2}V_{n_2,k_3,n_1,k_4}$$
$$\times\ \mathcal{G}_{n_1}(\tau'-\tau,x',x)\mathcal{G}_{n_2}(\tau-\tau',x,x')\,,$$
$$\mathcal{G}_k(\tau>0,x,x') = \sum_j \chi_j(x)\chi_j^*(x')\frac{e^{-\tau\left(\varepsilon_k+\omega(j+1/2)\right)}}{1-e^{-\beta\left(\varepsilon_k+\omega(j+1/2)\right)}} = \mathcal{G}_k(\tau-\beta,x,x')\,. \tag{92}$$

As $\varepsilon_k > \Lambda$ the Matsubara Green's function of the high energy channels is very short-ranged in both imaginary time and space, so that retardation effects can be neglected and working with a purely local interaction between the low-energy channels remains justified. Hence the quartic terms generate only a very small renormalization of the interaction terms already present between the low-energy channels.

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
