# Peer review of "Josephson oscillations in split one-dimensional Bose gases"

_SciPost Physics, doi:SciPost Phys. 10, 090 (2021)_

## Round 1 · Referee Report · Anonymous (Referee 1) · 2020-12-12

Strengths

1- Very experimentally motivated 2- Microscopic analysis of the experimental setup 3- Points out as, in this parametric range, previous descriptions in terms of the sine-Gordon model were insufficient 4- Clearly written

Weaknesses

1- Weak excitations/high-temperature regime 2- No quantitative comparison with experimental data 3- Corrections beyond the quadratic approximation are hard to be computed

Report

The authors pursue their study of an interesting experimental setup, namely two coupled 1d quasicondensates. In their precedent studies, they analyzed the putative low energy description of the low-energy sector of the model where, after bosonization, the phase difference should be described by the sine-Gordon (SG) field theory. This low energy description has been proven to be reliable at equilibrium (Ref. [11]), but its applicability in out-of-equilibrium setups is questionable. An out-of-equilibrium experimental protocol currently under investigation consists of a dynamical splitting of a single quasicondensate: the splitting process excites the system with the possibility of leaving the realm of applicability of the SG approximation. Indeed, previous attempts in describing the experiment based on the SG model (some of them from the same authors of this manuscript), failed to capture even qualitatively the experimental observations, primarily the phase-locking. This led the authors to question the whole SG approximation and analyze the protocol from a first-principle microscopic viewpoint, modeling the gas as an array of weakly interacting 1d systems, where different species are associated with the energy levels of the transverse trap. The interacting model is then addressed within a self-consistent Hartree-Fock approximation (SCHFA), which holds at short times and for weak interactions.

I think this work addresses the important question of correctly modeling an interesting experimental setup and the fact that this description qualitatively captures experimental observations, e.g. the phase-locking, which were not reproducible within the sine-Gordon approximation, points out as the latter is most likely not sufficient. I have some questions that I would like the authors to address.

  • I see at least two conceptually differences in this modeling when compared with the previous ones. The first point is dropping the low-energy description (bosonization) of the quasicondensate of the relative phase, the second is the inclusion of more excited states. I understand that both these features are needed for a satisfactory description of the experiment, but I wonder if a self-consistent approximation of a single species (i.e. the lowest transverse mode in Fig. 1), but evolved with the SCHFA without using bosonization, gives hints of phase-locking. With the current experimental parameters, all the first three levels are populated, but one could wonder what would happen at lower energies when only the first state matters.

  • Is there any simple explanation behind the closure of the gap between the second and third states in Fig. 1?

  • Splitting a single condensate has dramatic effects on the system, with the result of populating the excited transverse modes. On the other hand, at equilibrium, the two coupled quasicondensates are described by the SG model, suggesting that only the lower-energy transverse mode matters. I think that a natural question is: what does it happen out-of-equilibrium with changes in the potential barrier, but keeping it always finite? I would expect that, at least for small changes, the lowest-energy transverse mode should suffice in describing the dynamics. In that case, it could be that the experiments still acts as a quantum simulator of the SG field theory. Can the authors comment on this?

-I find Sec. 7 a bit superfluous. The limitations of the SCHFA are well understood and described in the previous sections and, as the same authors admit, the equations of Sec. 7 are essentially untractable and there is no hope to quantitatively describe the corrections to the SCHFA. I would like the authors either to further justify the importance of the content of Sec. 7, or to let it aside (or maybe move it to the appendixes).

Once my questions have been properly addressed, I can recommend this paper for publication in Scipost Physics.

Requested changes

See report

  • validity: high
  • significance: high
  • originality: high
  • clarity: top
  • formatting: perfect
  • grammar: perfect

Author:  Yuri Daniel van Nieuwkerk  on 2021-02-15  [id 1238]

(in reply to Report 1 on 2020-12-12)

Please see the attached pdf for our reply to Referee 1

Attachment:

reply_ref1.pdf

---

## Round 1 · Referee Report · Anonymous (Referee 2) · 2020-12-17

Report

Referee report

In this paper the authors develop and numerically implement a self-consistent time-dependent Hartree-Fock (SCHF) approach, which they use to study tunnel-coupled effectively 1D bosonic condensates with a focus on the out-of-equilibrium dynamics of the system in situations of direct experimental relevance [1, 2]. Performing a careful expansion of the field operators with respect to the transverse directions of the problem they obtain an effective model with 2-3 channels that is manageable within their approach even when the longitudinal spatial dependence of the trapping potential is taken into account. Starting from a thermal state, their approach enables the authors to simulate a rather non-trivial experimental protocol that prepares the system in a particular initial state from which the subsequent time evolution can be studied as well. One main finding of the work is the damping of the relative phase and particle number imbalance, which is in accordance with the experimental results, and which was so far not explained by other theoretical attempts [3, 4, 5]. Moreover, the qualitative dependence of the SCHF damping time on the total particle number also shows a similar behaviour to that of the experiments, although the temperature in the theoretical framework is larger and the particle number is smaller than in the real-world setup. An important comparison is made with an alternative scenario with a box-potential for the longitudinal direction, which shows no damping. Based on these results (and also relying on earlier theoretical approaches neglecting spatial inhomogeneities and reporting the absence of damping [3, 4, 5]), the authors come to the conclusion that inhomogeneous longitudinal trapping potential has an important role in the experimentally observed damping of various quantities.

The grammar and the style of the text is very good, and the paper is structured in a logical way. In general the paper is well-written, it is detailed enough and at the same time remains not difficult to follow. Discussing briefly how some relevant quantities are extracted from the experimental data is particularly appreciated. The paper cites the relevant works from the literature properly as well.

In this work several approximations are made and whenever possible the authors carefully justify these steps or comment on the circumstances under which they can be done. Perhaps the most important approximation is, nevertheless, the HF time evolution itself and estimating its error or rigorously justifying its applicability is a rather difficult problem. Therefore the authors first discuss how accurately the HF approximation can yield the initial density profile for the interacting Bose gas in an elongated single well potential. This quantity is compared with that of computed with the local density approximation and the comparison shows a nice match between the results for not too high atom numbers. A very important finding is that in SCHF framework, the monotonic dependence of the damping time (being qualitatively similar to the experimental results) breaks down around the same particle number, where the conventional HF method starts failing to accurately predict the atom density in the equilibrium case. I think this observation is a strong confirmation on the validity or at least on the consistency of the SCHF approach.

The results presented in the paper are new and interesting and significantly help clarify the physical mechanism leading to a quick damping to a phase-locked state in the experiments discussed [1, 2]. Based on the above mentioned comparison, I think that the reliability/consistency of the novel results is also properly justified. The ability of the SCHF method to simulate the experimental initialising protocol is also noteworthy. I therefore recommend this work for publication, but before giving the ultimate green light I would like to address a few minor questions to the authors. I kindly ask them to respond to these questions and based on their own discernment to consider the need for discussing some of these issues in their paper as well.

• Why are the simulations in section 6 stopped at 80-120 ms? Would it be possible to follow the time evolution further in time? If yes, that could make the damping more visible.

• It is an important feature of the experiment discussed [1, 2] and is already mentioned in the introduction of this paper, that not only a damping of the relative phase and the particle number imbalance is observed, but a relaxation to phase locked steady state. In other words, the fluctuations of the relative phase are also damped. In this paper $<\varphi(x,t)>$ is essentially obtained as $\arg\left[<\Psi_{R}^{\dagger}(x,t)\Psi_{L}(x,t)>\right]$. This latter quantity can be written as $\propto\arg\left[<e^{i\varphi(x,t)}>\right]$, hence the SCHF approach also predicts that the phase fluctuations are damped as well. I think this point might be worth emphasising.

• On Pg. 13, the last sentence reads: 'In particular, $\Delta_{2}$ shows whether the connected 4-point function, which is zero in HF, has a significant value in the initial state.' Based on the definition of $\Delta_{2}$ I think this sentence needs some more clarifications, I find it slightly misleading: $\Delta_{2}$ is the connected 4-point function directly only if $\Delta_{1}$ is zero. But certainly, even if $\Delta_{1}$ is small but non-zero, $\Delta_{2}$ is a good measure for the 4-point connected correlations.

And some minor (potential) typos

• In the caption of Fig. 4 and Fig.11, if I am not mistaken, the temperature T is 60nK rather than 60K.

References

[1] Marine Pigneur, Tarik Berrada, Marie Bonneau, Thorsten Schumm, Eugene Demler, Jörg Schmiedmayer, Relaxation to a Phase-locked Equilibrium State in a One-dimensional Bosonic Josephson Junction, Phys. Rev. Lett., 120, 173601 (2018).

[2] Marine Pigneur, Jörg Schmiedmayer, Analytical Pendulum Model for a Bosonic Josephson Junction, Phys. Rev. A 98, 063632 (2018).

[3] Yuri D. van Nieuwkerk, Fabian H. L. Essler, Self-consistent time-dependent harmonic approximation for the sine-Gordon model out of equilibrium, SciPost Phys. 5, 046 (2018).

[4] E.G. Dalla Torre, E. Demler and A. Polkovnikov, Universal Rephasing Dynamics after a Quantum Quench via Sudden Coupling of Two Initially Independent Condensates, Phys. Rev. Lett. 110 (2013) 090404.

[5] D. X. Horvath, I. Lovas, M. Kormos, G. Takacs, G. Zarand, Non-equilibrium time evolution and rephasing in the quantum sine-Gordon model, Phys. Rev. A 100, 013613 (2019).
  • validity: -
  • significance: -
  • originality: -
  • clarity: -
  • formatting: -
  • grammar: -

Author:  Yuri Daniel van Nieuwkerk  on 2021-02-15  [id 1237]

(in reply to Report 2 on 2020-12-17)
Category:
answer to question

Please see the attached pdf for our reply to Referee 2.

Attachment:

reply_ref2.pdf

---

## Round 2 · Referee Report · Anonymous (Referee 1) · 2021-2-17

Report

I have read the resubmitted version of the manuscript, together with the authors' reply to my previous report.
I am fully satisfied with the authors' answers and changes, therefore I can recommend this paper for publication.

---

## Round 2 · Referee Report · Anonymous (Referee 2) · 2021-2-24

Report

I am grateful for the authors’ clarifications and find the answers to my questions, as well as the modification in the paper adequate. I therefore recommend this work for publication.

---

## Round 2 · Author Response

We thank the referees for their very helpful comments. We have responded to their questions and comments in a point-by-point manner in the pdf documents uploaded at the submission page.

---

## Round 2 · List of Changes

-Expanded details of the transverse potential in Sec. 5.1 (Eqs. 54 and 55)
-Added a comment about the reemergence of the oscillations and the relevant time scales (bottom p. 20)
-Added a comment about the time scale of the damping vs the time scale of the gases' breathing motion (top of p. 22).
-Added a comment explaining the relevance of the equations that go beyond SCHF in Sec. 7.
-Elaborated on the useful applications of our work to the modelling of the experiments using (perturbations of) a sine-Gordon model.

---

## Editorial Decision

published